# Inflammatory cells dynamics control neovascularization and tissue healing after localized radiation induced injury in mice

Céline Loinard [1✉], Mohamed Amine Benadjaoud[1], Bruno Lhomme[1], Stéphane Flamant [1], Jan Baijer[2] & Radia Tamarat[1]

Local overexposure to ionizing radiation leads to chronic inflammation, vascular damage and cachexia. Here we investigate the kinetics of inflammatory cells from day (D)1 to D180 after mouse hindlimb irradiation and analyze the role of monocyte (Mo) subsets in tissue revascularization. At D1, we find that Mo and T cells are mobilized from spleen and bone marrow to the blood. New vessel formation during early phase, as demonstrated by ~1.4- and 2-fold increased angiographic score and capillary density, respectively, correlates with an increase of circulating T cells, and Mo$^{hi}$ and type 1-like macrophages in irradiated muscle. At D90 vascular rarefaction and cachexia are observed, associated with decreased numbers of circulating Mo$^{lo}$ and Type 2-like macrophages in irradiated tissue. Moreover, CCR2- and CX3CR1-deficency negatively influences neovascularization. However adoptive transfer of Mo$^{hi}$ enhances vessel growth. Our data demonstrate the radiation-induced dynamic inflammatory waves and the major role of inflammatory cells in neovascularization.

[1] Institut de Radioprotection et de Sûreté Nucléaire (IRSN), Fontenay-aux-Roses, France. [2] CEA, Fontenay-aux-Roses, France. ✉email: celine.loinard@irsn.fr

High-dose acute radiation accidents occur each year, following industrial accidents (loss of radioactive sources) or overdosage following medical applications (radiotherapy and interventional radiology procedures). The radiation injury is characterized by successive and unpredictable inflammatory waves over the first few days to years after irradiation, and these lead to the horizontal and vertical extension of tissue injury, including vascular rarefaction and muscle cachexia[1]. Vascular damage and rarefaction are considered to be the main cause of long-term morbimortality in patients leading to tissue ischemia after ionizing radiation exposure[2]. Local overexposure to ionizing radiation has severe health consequences, especially when the absorbed dose exceeds 25 Gy and leads to tissue necrosis[1,3]. Repair of acute skeletal muscle injury is a tightly regulated process, which mainly consists in three phases, including inflammation, regeneration, and angiogenesis[4,5]. In the preclinical model of total body irradiation, it has been shown that the numbers of both myonuclei and satellite cells per myofiber were decreased in a dose-dependent manner[6]. Moreover, a single irradiation dose of 18 Gy blocks muscle regeneration by inducing lethality of myoblasts[7]. Studies of muscular pathology using more than 25 Gy irradiation dose have shown morphological alterations, hemorrhage, necrosis, inflammation, fibrosis and mitochondria destruction[8–11]. Ischemia is the common process in both diseases radio-local injury and cardiovascular disease. In ischemic disease, insufficient organ perfusion following thrombotic vessel obstruction of the feeding artery is a major determinant of postischemic remodeling[12]. However, exposure of mammalian cells such as endothelial cells to ionizing radiation leads primarily to DNA damage-induced cell death[13]. Ischemia is characterized by vascular damage/rarefaction and inflammation resulting in fibrosis characterized by collagen-based scar[14]. The ischemic tissue response is based on four principal processes, vasculogenesis, angiogenesis, arteriogenesis, and collateral growth, which contribute to tissue repair and remodeling during acute and chronic ischemic vascular diseases[15]. These processes result from hemodynamical forces changes within the vascular wall leading to modification of the vascular homeostasis[15,16].

Infiltration of inflammatory cells in hypoxic areas is a hallmark of tissue ischemia, and the respective role of distinct leukocyte subsets in postischemic neovascularization—CD4$^+$ and CD8$^+$ T cells[17,18], NK cells[19], regulatory T cells[20], mast cells[21], monocytes/macrophages[22]—is not completely understood. In particular, T lymphocytes are involved in this process, as demonstrated by the fact that nude mice, which lack all T-cell subsets, exhibit a pronounced reduction in postischemic vessel growth[23]. Leukocytes and monocytes trigger neovascularization through the release of several angiogenic/arteriogenic factors, including vascular endothelial growth factor (VEGF), pro-inflammatory cytokines such as tumor necrosis factor-α, interleukin (IL)-1β, and metalloproteinases[24,25]. In addition, the role of monocytes in the neovascularization process has been documented by different groups[18,26].

Monocytes are a heterogeneous population with two major subtypes in mice: Ly6C$^{hi}$Ly6G$^-$/4$^{hi}$ "inflammatory" monocytes (Mo$^{hi}$) and Ly6C$^{lo}$Ly6G$^-$/4$^{lo}$ "resident" monocytes (Mo$^{lo}$) corresponding to the CD14$^{hi}$CD16$^-$ and CD14$^{lo}$CD16$^+$ subpopulations in humans, respectively[27]. Mo$^{hi}$ express the inducible NO synthase (NOS) and pro-inflammatory cytokines, such as IL-1 and IL-12, whereas the Mo$^{lo}$ produce large amounts of arginase 1, the anti-inflammatory cytokine IL-10, and VEGF. Mo$^{hi}$ rapidly enter sites of inflammation, while Mo$^{lo}$ enter lymphoid and non-lymphoid organs under homeostatic conditions, patrol across the vascular endothelium in a CX3CR1-dependent manner[28] and favor tissue regeneration in diverse contexts[29]. We recently showed the central role for monocyte/macrophage activation in the orchestration of neovascularization mechanism in a mouse model of local colorectal irradiation[22]. In addition, in a mouse

model of hindlimb ischemia, the transplantation of Mo$^{hi}$, but not of Mo$^{lo}$, led to the activation of postischemic neovascularization[26].

Monocyte recruitment to ischemic areas is thought to occur mainly via chemokine/chemokine receptor signaling. In particular, CCL2 and its cognate receptor CCR2, as well as fractalkine (CX3CL1) and the ligands of CX3CR1 are implicated in this context[30–32]. In fact, *CCL2* is upregulated at site of collateral growth, and revascularization is markedly enhanced with *CCL2* treatment[33]. Similarly, in ischemic hindlimb models, deficiency in *CCL2* or *CCR2*, reduces postischemic inflammation and vessel growth[25,34] and enhances gastrocnemius muscle atrophy[35]. Therapeutic modulation of the inflammatory response may therefore hold promise to improve reparative response for the prevention of postischemic disease[15,16].

Here, using a mouse model of localized hindlimb radiation-induced injury, we analyzed the dynamics of mobilization and recruitment of innate and adaptive inflammatory cells in different tissues (bone marrow (BM), spleen, blood, and muscles) from D1 to D180 post irradiation. We showed for the first time the inflammatory waves characterizing ionizing radiation injury. These waves corresponded to the alternance between proliferation/mobilization phases observed in different tissues, such as the spleen, blood, and muscle; however, they were less pronounced in BM. We highlighted two phases after hindlimb irradiation, corresponding first to an early phase with inflammation and neovascularization correlated with the mobilization of CD4- and CD8 T cells from the spleen, infiltration in the muscle of Mo$^{hi}$ and their differentiation into pro-inflammatory macrophages (Type 1-like macrophages/M1-like). In the late phase, we demonstrated vascular rarefaction and cachexia, which correlated with a decrease of Mo$^{lo}$ counts in blood and muscle and the differentiation of Mo$^{lo}$ into Type 2-like macrophages (M2-like) in muscle.

Finally, we investigated more precisely the role of Mo$^{hi}$ and M1-like on neovascularization during the early phase. The deficiency on *CCR2* or *CX3CR1* hampered arteriogenesis mechanisms, whereas Mo$^{hi}$ intramuscular injection in WT mice improved arteriogenesis.

## Results

**Localized hindlimb irradiation induces injury and rhabdomyolysis.** To investigate the irradiation effect on the hindlimb, we first assessed the kinetics of evolution of the lesion by a semi-quantitative analysis of the wound extent, ulceration, moist desquamation, and limb retraction during the wound healing process each week until D180. After 14 days post irradiation, the injury score increased and peaked at D35 and then slowly decreased until D180 (Fig. 1a). This deleterious effect of irradiation was associated with cachexia/muscle wasting (rhabdomyolysis). Gastrocnemius and tibialis weight decreased progressively and significantly at D120, 150, and 180 by 2- and 1.5-fold, respectively, compared with nonirradiated group ($P < 0.01$, Fig. 1b).

**Localized hindlimb irradiation induces proliferation, mobilization, and recruitment of innate inflammatory cells.** BM and spleen are known to release inflammatory cells after injury and contribute to the inflammatory process after irradiation[22]. CD11b$^+$ leukocytes corresponding to neutrophils and monocytes were mobilized from the BM or spleen to the blood and then infiltrated in irradiated muscle (Fig. 2a, b). We did not see any accumulation of monocytes in muscle, suggesting they differentiated in macrophages[36]. Nonirradiated (NIR) corresponds to littermates euthanized at different time points and showed comparable results in term of the number of inflammatory cells in tissues (blood, spleen, bone marrow, and muscles) at all times.

In BM, neutrophils (CD11b$^+$Ly6G$^-$/4$^{hi}$) significantly proliferated at D1, D150, and D180 ($P < 0.001$) compared to NIR

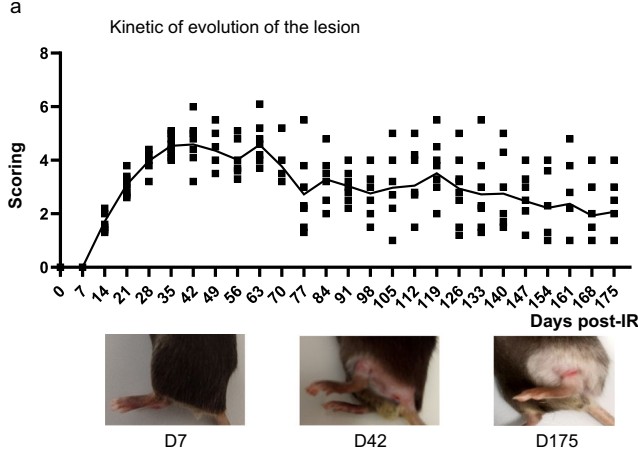

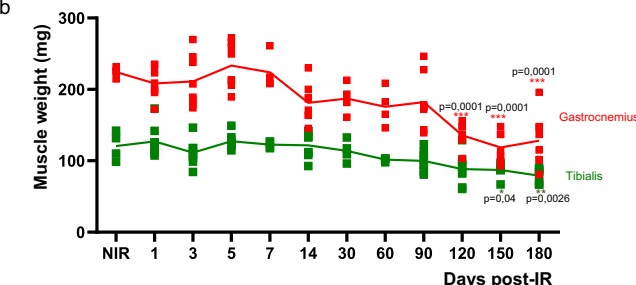

**Fig. 1 Hindlimb irradiation-induced injury and muscular waste. a** Kinetic evolution of injury scores mesured each week and representative photographs of radiation-induced hindlimb injuries at different time points. **b** Evolution of muscular weight measured on tibialis and gastrocnemius, nonirradiated (NIR) and irradiated, at days 1, 3, 5, 7, 14, 30, 60, 90, 120, 150, and 180 post irradiation. *$P < 0.05$; **$P < 0.01$; ***$P < 0.001$ vs. NIR; $n = 8$ animals /time point, ANOVA $+/-$ SEM.

(Fig. 2a). Mo$^{hi}$ (CD11b$^+$Ly6G-7/4$^{hi}$) proliferated at D14 and D180 ($P < 0.05$) and Mo$^{lo}$ (CD11b$^+$Ly6G-7/4$^{lo}$) at D150 and D180 ($P < 0.001$) respectively compared to NIR (Fig. 2a). Interestingly, at D5, neutrophils, Mo$^{hi}$ and Mo$^{lo}$ were depleted compared to NIR and then proliferated to return to basal level at D7 until D90. In addition, neutrophils, Mo$^{hi}$, and Mo$^{lo}$ numbers increased in the spleen at D14 and D150 ($P < 0.001$, Fig. 2b).

In the blood, neutrophils, Mo$^{hi}$, and Mo$^{lo}$ counts increased from D1 until the end of the experiment ($P < 0.001$, Fig. 2c).

In muscle, Mo$^{hi}$, Mo$^{lo}$, and neutrophils were recruited and peaked at D1, D5, and D60 and then returned to basal level at D3, D30, and D120 (Fig. 2d). M2-like peaked at D1 ($P < 0.01$, Fig. 2e), whereas dendritic cells peaked at D60 ($P < 0.01$, Fig. 2e) after muscle irradiation. We also noticed an increase, though not significant, of M1-like in muscle at D60.

These results suggest that, after muscle irradiation, innate inflammatory cells are recruited early and massively from spleen and BM into the muscle. Moreover, these results show the time-recurring, consecutive inflammatory waves following radiation injury, corresponding to alternate phases of innate inflammatory cells proliferation in BM and spleen (Fig. 2a, b) and mobilization through the blood (Fig. 2c), and then their infiltration in irradiated muscle (Fig. 2d, e).

**Localized hindlimb irradiation induces the mobilization and recruitment of adaptive inflammatory cells.** We previously showed, in a mouse model of colorectal injury, that after

irradiation, T lymphocytes were recruited to the injured tissue[22]. Here we found that, in our model, CD4 and CD8 T cells were similarly mobilized from the spleen and showed a strong significant decrease from D1 ($P < 0.001$, Fig. 3a) until D90. Afterward, T cells proliferated and almost reached basal level at D120, and then T-cell numbers decreased again at D150 and D180 ($P < 0.001$, Fig. 3a) compared to NIR. In blood, T-cell numbers increased and peaked at D30 and D60 ($P < 0.001$, Fig. 3b). In muscle, T cells were recruited and peaked at D1 and D30 ($P < 0.001$, Fig. 3c).

These results show that localized hindlimb irradiation induced a depletion of T-cell numbers in the spleen, suggesting their mobilization and recruitment from the spleen to the irradiated muscle. Similar to innate cells, we observed consecutive waves of T-cell mobilization and recruitment in blood and muscles. However, we did not observe such waves in the spleen, probably because the T cells were immediately mobilized.

**Localized hindlimb irradiation induces cytokines and chemokines expression in muscle.** *CCL2* and *CX3CL1* are known to contribute to neovascularization via the attraction and retention of monocytes and T cells in ischemic leg[26,37].

mRNA expression of *CCL2* significantly increased by 11- and 14-fold at D14 and D30, respectively, in irradiated muscle compared to NIR (Fig. 4a). Similarly, mRNA expression of *CX3CL1* significantly increased by 3.7-fold at D30 and 7.7-fold at D90 in irradiated muscle compared to NIR tissue (Fig. 4b).

Chemokine upregulation was associated with macrophage and lymphocyte infiltration in muscle. At D7 and D14 after irradiation, the number of CD68-positive macrophages increased by 3.5- and 4-fold ($P < 0.001$), respectively, and decreased at D30, compared to NIR tissue. The number of CD3-positive lymphocytes increased by 39-fold at D7, 24–fold at D14, and 18-fold at D30 (Fig. 4b, c and supplementary Fig. 1).

These results show that localized irradiation results in the upregulation of chemokines such as *CCL2* and *CX3CL1* and inflammatory cell recruitment.

**Localized hindlimb irradiation induces transient neovascularization process in muscle.** We next analyzed the pro-angiogenic actors *eNOS* in muscle post irradiation. *eNOS* mRNA levels in irradiated muscle were upregulated by 2.8-, 2.6-, 2.5-, and 2.3-fold, respectively, at D7, D14, D30, and D90 compared to NIR (Fig. 5a). Accordingly, *eNOS* protein levels were also significantly upregulated at D30 by 2.3-fold compared to NIR ($P < 0.001$, Fig. 5b).

Furthermore, we evaluated the neovascularization process following irradiation in the muscle by using independent and complementary microangiography and immunohistochemistry approaches. The angiographic score was significantly increased by 1.4-fold ($P < 0.01$) at D14 compared with the NIR group. In addition, capillary density was upregulated at D7 and D14 by 1.7- ($P < 0.05$) and twofold ($P < 0.001$), respectively, compared with the NIR group (Fig. 5c, d). Moreover, by the Bayesian latent variable approach[38], we analyzed the correlation between neovascularization process and mobilization/recruitment of inflammatory cells during the early phase (D0-D30) and late phase (D60-D180). From D1 to D30 post irradiation, we found a significant correlation between new vessel formation and mobilization of CD4/CD8 cells ($P < 0.01$ both) from the spleen and their increase in the blood ($P < 0.05$ both) (Table 1). In the same way, the increase of Mo$^{hi}$ infiltrated numbers and their differentiation into M1-like in irradiated muscle was associated with the neovascularization ($P < 0.01$ both) (Table 1).

However, at D60, angiographic score returned to the basal level, and then decreased significantly at D90, D120 and D150

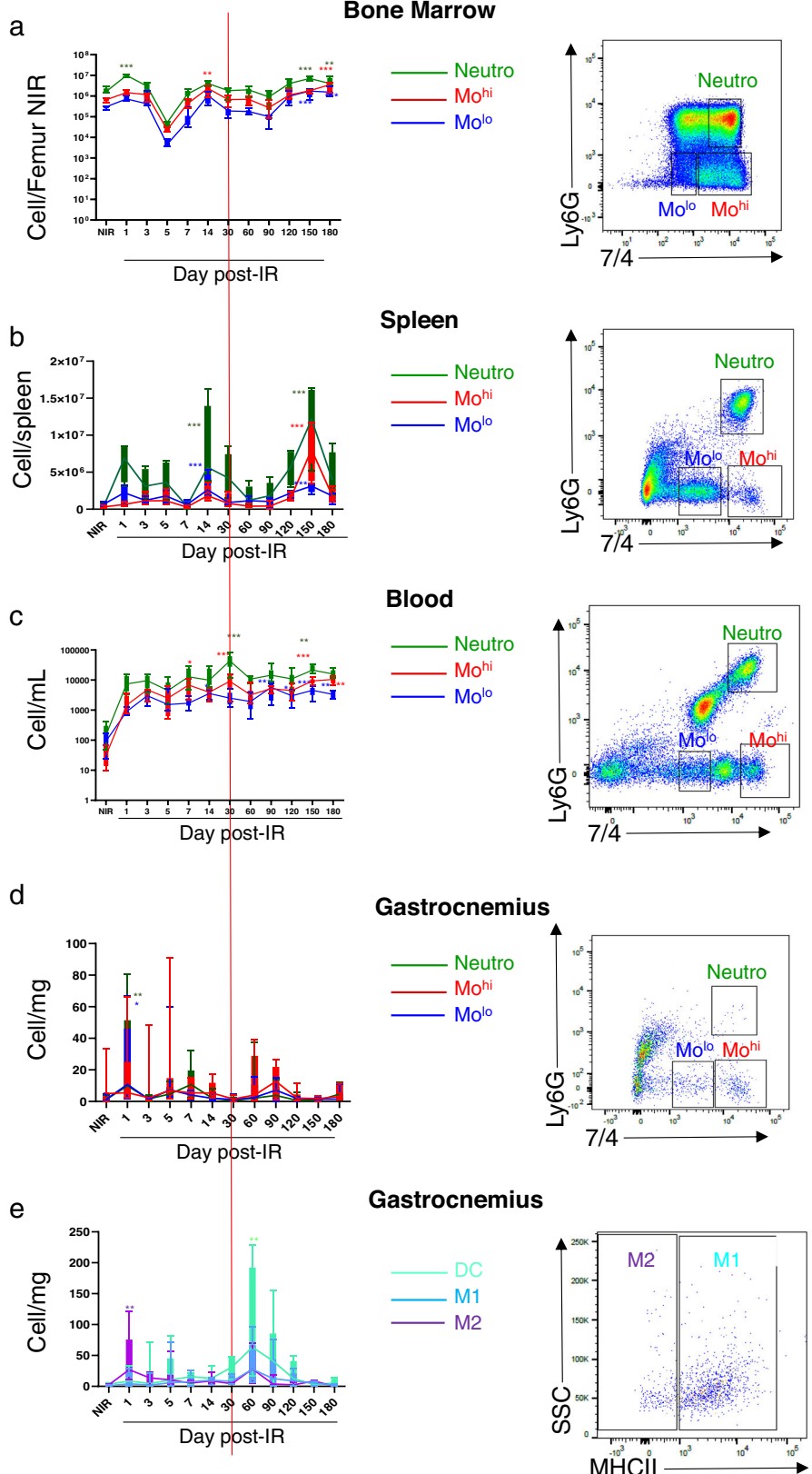

**Fig. 2 Mobilization and recruitment of innate inflammatory cells. a–d** Quantitative analysis (left) and representative flow cytometry gating (right) of cell counts for neutrophils (CD45 + CD11b+Ly6G+7/4hi, green line), Mohi (CD45 + CD11b+Ly6G-7/4hi, red line) and Molo (CD45 + CD11b+Ly6G-7/4lo, blue line) (gated on CD45+ and CD11b+ cells) in bone marrow, spleen, blood, and muscle, respectively. **e** Quantitative analysis (left) and representative flow cytometry gating (right) of cell counts for DC (CD45 + CD11b+CD11c+MHCII+), M1-like Mϕ (CD45+CD11b-F4/80+CD64+MHCII+) and M2-like Mϕ (CD45+CD11b-F4/80+CD64+MHCII−) in muscle, nonirradiated (NIR) and irradiated, at days 1, 3, 5, 7, 14, 30, 60, 90, 120, 150, and 180 post irradiation. *P < 0.05; **P < 0.01; ***P < 0.001 vs. NIR; n = 8–10 animals /time point, ANOVA +/− SEM.

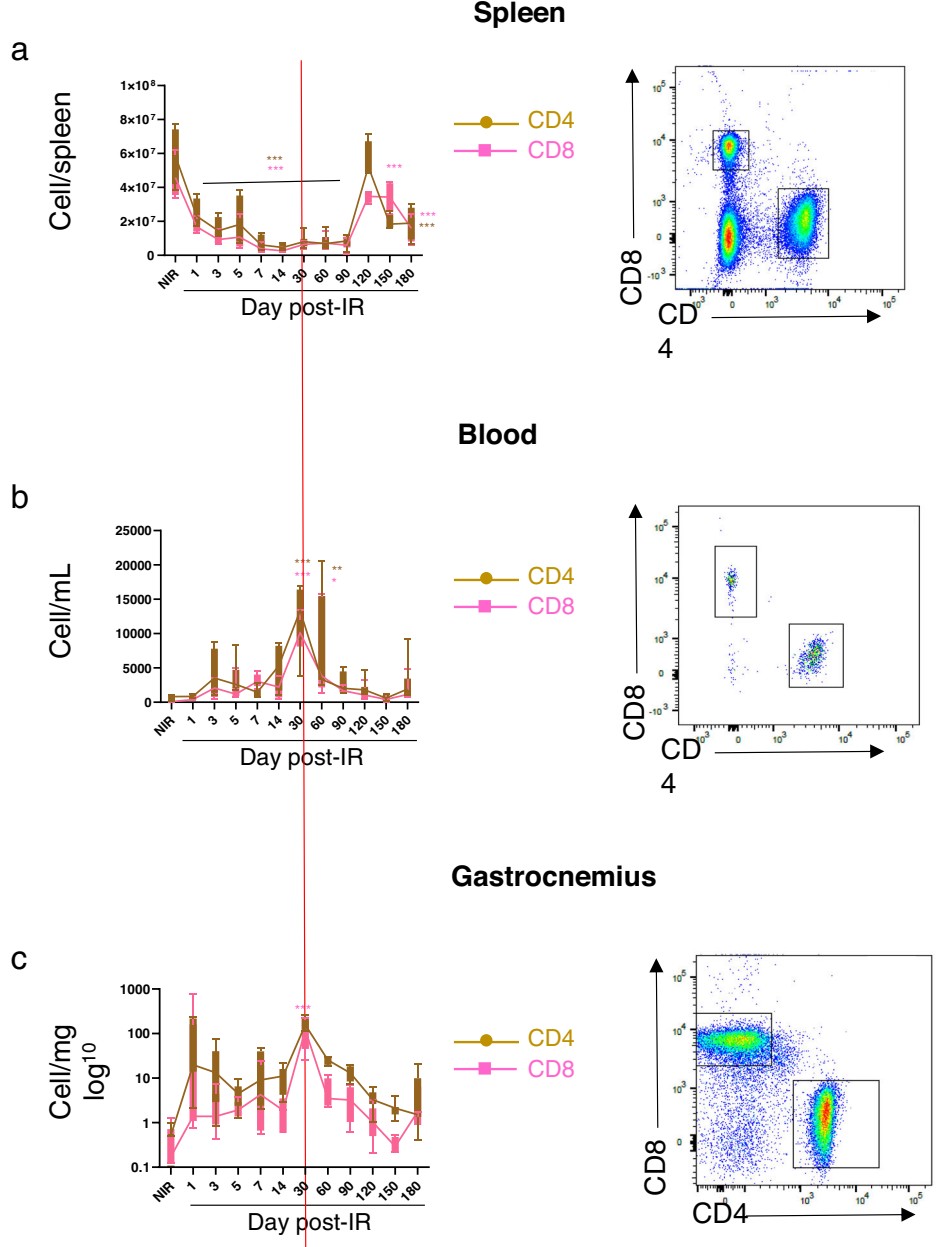

**Fig. 3 Mobilization and recruitment of adaptative inflammatory cells. a–c** Quantitative analysis (left) and representative flow cytometry gating (right) of cell counts for CD4 (CD45+CD3+CD4+) and CD8 (CD45+CD3+CD8+) in spleen, blood, and muscle, nonirradiated (NIR) and irradiated, at days 1, 3, 5, 7, 14, 30, 60, 90, 120, 150, and 180 post irradiation. *$P < 0.05$; **$P < 0.01$; ***$P < 0.001$ vs. NIR; $n = 8$–10 animals/time point, ANOVA $+/-$ SEM.

($P < 0.01$) compared with the NIR group (Fig. 5c). Furthermore, at D30, capillary density returned to the NIR level and significantly decreased from D60 to D180 ($P < 0.05$) compared to the NIR group (Fig. 5d). More importantly, from D60 to D180, we demonstrated by Bayesian approach that the decrease of Mo$^{lo}$ counts in blood ($P < 0.05$) and in muscle ($P < 0.001$), as well as their differentiation in M2-like in muscle ($P < 0.05$) correlated with this process of vascular rarefaction (Table 1).

**Differential role of chemokine receptor in the control of circulating monocyte levels after irradiation.** In order to directly address the role of monocytes/macrophages in the neovascularization process after hindlimb irradiation, we investigated more precisely the role of Mo$^{hi}$ and Mo$^{lo}$ by using *CCR2*- or *CX3CR1*-deficient mice. In *Ccr2*$^{-/-}$ mice, circulating Mo$^{hi}$ ($P < 0.001$, Fig. 6a) and Mo$^{lo}$ ($P < 0.01$, Fig. 6b) levels were significantly

reduced when compared with wild-type mice at D3 following irradiation. Moreover, Lu et al. showed that mobilization of monocytes (Mo$^{hi}$ and Mo$^{lo}$), but not lymphocytes or neutrophils, was impaired from BM to blood and from blood to injured muscle in *Ccr2*$^{-/-}$ mice suggesting that the low level of monocyte analyzed in blood was not due to hindlimb irradiation[39]. Similarly, we also demonstrated the low number of macrophages in *CCR2*$^{-/-}$ muscle compared to WT mice at 10 days post irradiation ($P < 0.05$; Fig. 6d). Next, we sought to evaluate the impact of the *CCR2* and *CX3CR1* signaling pathways on postirradiation vessel growth. Ten days after irradiation, angiographic score was reduced in CCR2- and CX3CR1-deficient mice compared to their wild-type counterparts ($P < 0.001$, Fig. 6e).

As expected, our results clearly show that disruption of the *CCL2/CCR2 and CX3CL1/CX3CR1* pathways hampers postischemic vessel growth.

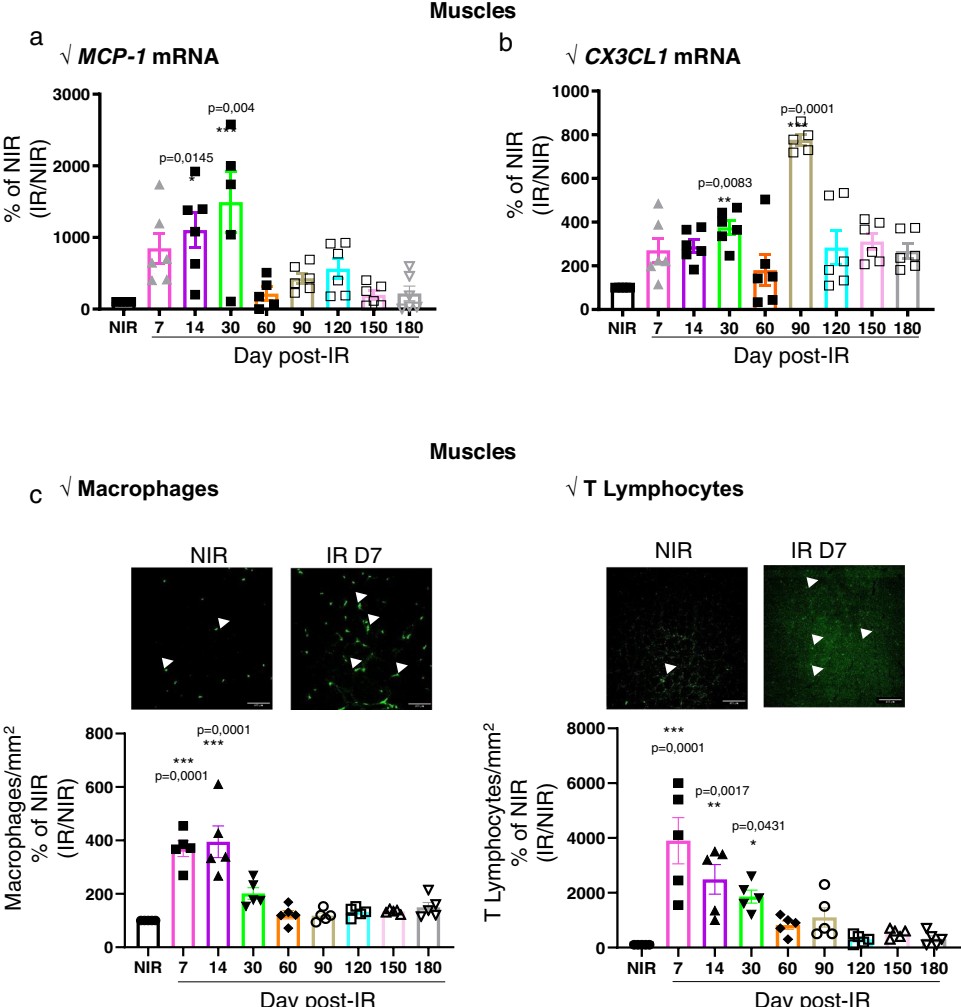

**Fig. 4 Radiation induces chemokine gene expression and recruitment of leukocytes. a** Quantitative evaluation of *CCL2* and *CX3CL1* mRNA in muscle. **b**, **c** Quantitative evaluation (left-side panels) and representative photomicrographs (right-side panels) of CD68-positive cells (**b**, indicated by arrows) or CD3-positive cells (**c**, indicated by arrows) per mm² in the nonirradiated (NIR) and irradiated muscle. Results were expressed as percentages relative to NIR, set to 100% and irradiated muscle was normalized to the NIR muscle at days 1, 3, 5, 7, 14, 30, 60, 90, 120, 150, and 180 post irradiation, scale bar: 100 μm. *P < 0.05; **P < 0.01; ***P < 0.001 vs. NIR; n = 6 animals/time point, ANOVA +/− SEM.

**Mo^hi promote neovascularization after hindlimb irradiation.** Because Mo^lo monocytes are scarce in the BM and difficult to collect in sufficient amounts from blood, we used splenic Mo^hi and Mo^lo as surrogates for circulating monocyte subsets. In addition, it was shown that blood and spleen monocyte subsets are almost identical in transcriptome and function[40]. On the day after hindlimb irradiation, WT C57BL/6 mice received intramuscular injection of PBS, 10^5 Mo^hi, or 10^5 Mo^lo. 10 days following irradiation, Mo^hi transfer improved angiographic score (*P* < 0.5, Fig. 6f) compared with PBS-treated mice. In contrast, injection of Mo^lo did not improve angiographic score.

These findings clearly identify a central role for Mo^hi activation in the orchestration of vascular inflammation and the promotion of neovascularization in the acute phase after hindlimb irradiation.

## Discussion

Local overexposure to ionizing radiation has severe health consequences, especially when the absorbed dose exceeds 25 Gy and leads to tissue necrosis[1]. The radiation injury is characterized by successive and unpredictable inflammatory waves over the first few days to weeks after irradiation, and these lead to surface and depth extension of tissue injury including vascular rarefaction and muscle cachexia[1]. In this report, we showed, for the first time, the dynamic inflammatory waves characterizing radiation-induced injury. In the spleen and the bone marrow, these waves corresponded to alternate phases between cell proliferation and mobilization through the blood and then their infiltration in irradiated muscles. Theses waves were much more pronounced for neutrophils, Mo^hi and then Mo^lo mainly in the spleen and muscle. However, in the blood, the number of innate circulating cells were increased from D1 to D180 post irradiation, whereas in irradiated muscle, their number was quite low suggesting an accumulation in the blood. In addition, T cells waves were observed in blood and muscle. Many studies have already shown the mobilization and recruitment of inflammatory cells after colorectal localized irradiation[22], femoral ischemia or myocardial infarction[20,24,41]. However, compared to the classical cardiovascular ischemic model[14], the numbers of M2-like macrophages in irradiated muscle were increased only at D1. We have several hypotheses that could explain this observation, including (i) the proliferation of M2 resident cells in the tissue, (ii) the defect in M2 egress (migration capability)[42] or (iii) Resistance of radio-induced apoptosis of M2 cells process in the muscle.

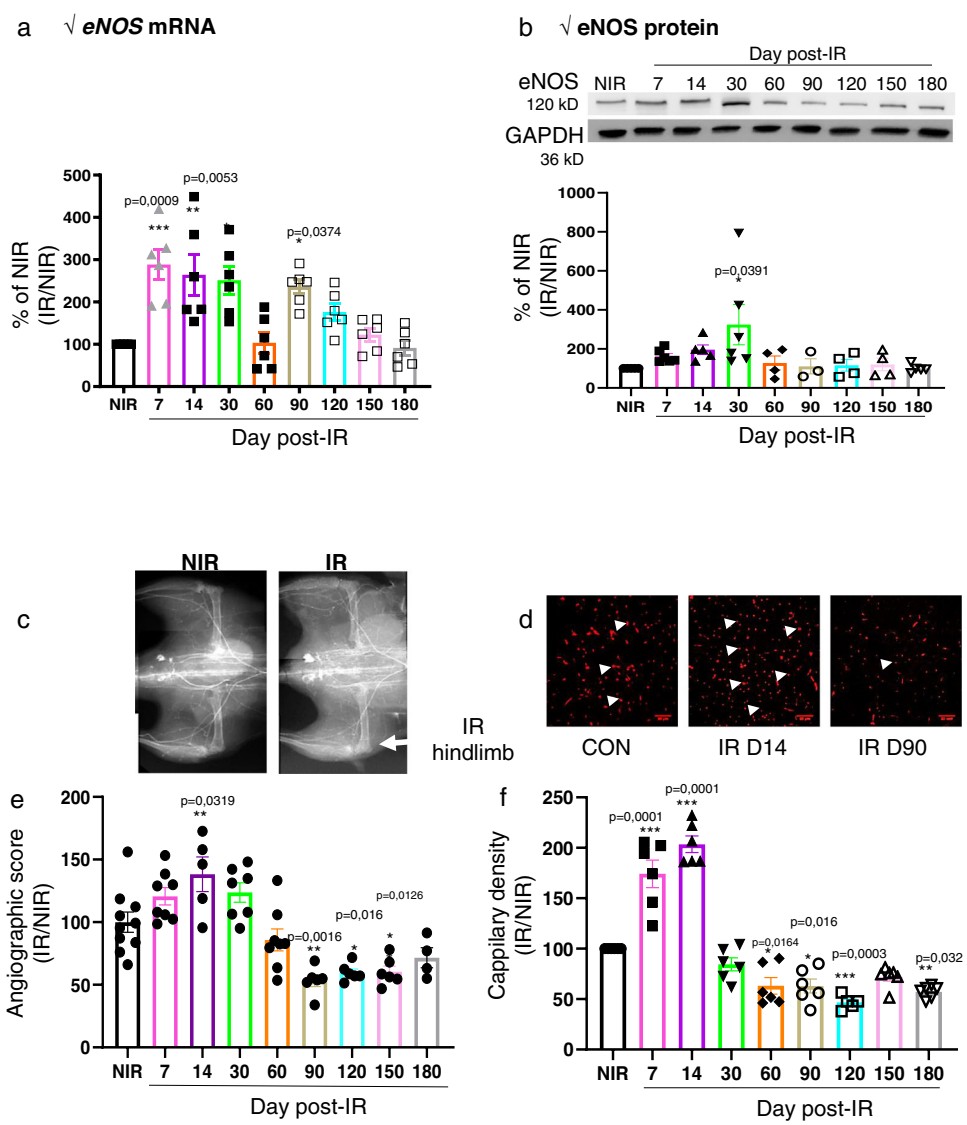

**Fig. 5 Radiation induces pro-angiogenic factors and transient neovascularization. a–d** Quantification of *eNOS* mRNA and protein levels in muscle. **e, f** Quantitative evaluation and representative photomicrographs of microangiography (**c**; vessels appear in white) and capillary density (**d**; capillaries appear in red, arrows indicate examples of isolectin B4-positive capillaries, scale bar: 100 μm). Results were expressed as percentages relative to nonirradiated (NIR), set to 100%, and irradiated muscle was normalized to NIR muscle, at days 1, 3, 5, 7, 14, 30, 60, 90, 120, 150, and 180 post irradiation. *$P < 0.05$; **$P < 0.01$; ***$P < 0.001$ vs. NIR; $n = 5$–8 animals /time point, ANOVA $+/-$ SEM.

**Table 1 Correlation between inflammatory cells and neovascularization.**

| D0-D30 post irradiation | Trend | Neovascularization | Association *P* values |
|---|---|---|---|
| Spleen-CD4 T cells | ↓ | ↑ | 0,002993589 |
| Spleen-CD8 T cells | ↓ | ↑ | 0,00430548 |
| Blood-CD8 T cells | ↑ | ↑ | 0,013917007 |
| Blood-CD4 T cells | ↑ | ↑ | 0,023232196 |
| Blood-Mo[hi] | ↑ | ↑ | 0,027728673 |
| Muscle-Mo[hi] | ↑ | ↑ | 0,004023073 |
| Muscle-M1-like | ↑ | ↑ | 0,009643325 |
| D60-D180 post irradiation | | | |
| Blood-Mo[lo] | ↓ | ↓ | 0,027514019 |
| Muscle-Mo[lo] | ↓ | ↓ | 0,004803924 |
| Muscle-M2-like | ↓ | ↓ | 0,014869191 |

Correlation analysis between inflammatory cells kinetics in the tissues (spleen, muscle, blood) and their role on angiographic score during the early phase from D0 to D30 and the late phase from D60 to D180 following hindlimb irradiation.

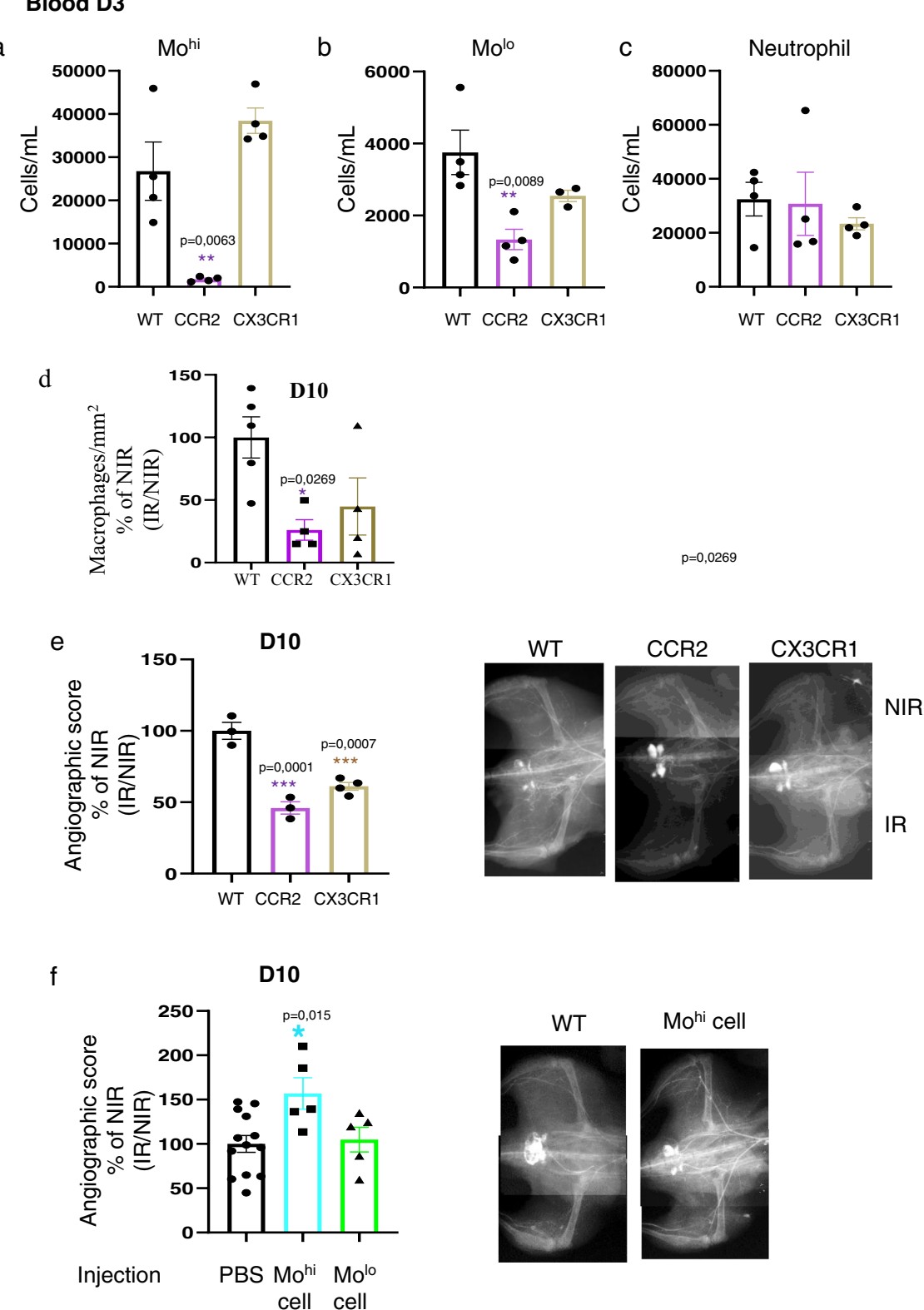

**Fig. 6 Mo^hi monocytes control and promote neovascularization after hindlimb irradiation. a–c** Quantitative analysis of cell counts for Mo^hi, Mo^lo, and neutrophils in blood in WT mice or CCR2- and CX3CR1-deficient mice. **P < 0.01; ***P < 0.001 vs. irradiated WT, n = 3–4 animals/time point. **d** Quantitative analysis of CD68+ macrophages infiltration from blood to injured muscles at D10 post irradiation in WT mice or CCR2- and CX3CR1-deficient mice, n = 5 animals/time point. **e** Quantitative evaluation and representative photomicrographs of microangiography in WT mice or CCR2- and CX3CR1-deficient mice 10 days after hindlimb irradiation. **f** Quantitative evaluation and representative photomicrographs of microangiography 10 days after hindlimb irradiation in animals intramuscular injected with PBS, 10^5 Mo^hi or 10^5 Mo^lo. *P < 0.05 vs. PBS injected mice. Results were expressed as percentage relative to nonirradiated (NIR), set to 100% and irradiated muscle was normalized to NIR muscle, n = 5–12 animals/time point, ANOVA +/− SEM.

Several studies have previously shown that irradiated ischemic endothelium displays pro-inflammatory properties. Endothelial cells express chemokines such as *CCL2* and *CX3CL1*, leading to the recruitment and infiltration of inflammatory cells through their receptors *CCR2* or *CX3CR1* to improve new vessel formation[26,30]. We showed that irradiated tissue expresses, during the early phase (D0-D30 post-IR), pro-angiogenic (*eNOS*) and pro-arteriogenic factors (*CX3CL1, CCL2*) to induce mobilization of pro-inflammatory CD4-, CD8 T cells, and Mo[hi] from the spleen and BM through the blood and their recruitment and infiltration in irradiated tissue leading to the differentiation of Mo[hi] into M1-like macrophages. Afterward, we observed a late phase (D60-D180 post-IR) characterized by a correlation between the reduced anti-inflammatory Mo[lo] recruitment and their differentiation into M2-like, vascular rarefaction, and cachexia. Although, at D90 post irradiation, we could detect an increase of *CX3CL1* and *eNOS* mRNA expression, we did not observe similar increase of the corresponding proteins, which is in agreement with vascular rarefaction and cachexia.

These results are in line with the literature concerning the role of *CCL2* and *CX3CL1* in participating to the recruitment of inflammatory cells and improving neovascularization. Hence after femoral artery occlusion, local infusion or overexpression of *CCL2* increased circulating monocyte numbers, and their accumulation was shown to support collateral and peripheral conductance, and neovascularization[26,31]. CX3CL1-deficient mice contained significantly fewer macrophages than controls in atherogenesis model[43]. These data add to accumulating evidence showing that monocytes/chemokines pathways play a central role in regulating angiogenesis. Moreover, T lymphocytes are also involved in this process since nude mice, which lack all T-cell subsets, exhibit a pronounced reduction in postischemic vessel growth[23]. Furthermore, CD4 and CD8 T-cell subsets have been suggested to play a role in vascular remodeling. Indeed, CD4- and CD8-deficient mice exhibit a significant reduction in post-ischemic vessel growth[17,18].

Next, we showed that monocytes contributed to neovascularization process after a localized hindlimb irradiation by using CCR2- and CX3CR1-deficient mice. The deletion of CCR2 or CX3CR1 reduced the number of circulating monocytes and decreased angiographic score as previously reported. CCR2 and CX3CR1 have been shown to control monocyte recruitment to ischemic tissues[30,44]. CCR2 is mainly involved in the regulation of circulating Mo[hi] levels. Of interest, lack of *CCR2* specifically abrogates Mo[hi] infiltration in the ischemic myocardium[29,40]. Also, it has been shown that *CCR2* expression by BM cells, but not by injured muscle tissue cells, is required to mount an inflammatory response following acute skeletal muscle injury[45] to allow subsequent muscle capillary arterialization[46]. Mo[hi] facilitate arteriogenesis and angiogenesis by producing growth factors[35,47–49]. Moreover, *CX3CR1* signaling has been shown to promote the survival of monocytes that are short-lived cells[50]: endogenous *CX3CR1* signaling might be required for the survival of newly mobilized BM-originating monocytes. This is supported by our observation of a lower number of Mo[lo] monocytes in CX3CR1[-/-] mice compared to WT animals. CX3CR1-deficient mice revealed adequate monocyte recruitment and revascularization for skin repair and wound angiogenesis; however, the myeloid cell response and magnitude of neovascularization were dampened compared with wild-type mice[51]. In addition, in atherosclerosis, many studies have shown that *CX3CL1/CX3CR1* signaling was implicated in angiogenesis during plaque microvessel formation[44,52,53].

Finally, we highlighted a major role of Mo[hi] monocytes in postischemic vessel growth. Muscular administration of Mo[hi] enhanced arteriogenesis, whereas Mo[lo] injection had no effect on this process. We show that adoptive transfer of the inflammatory subset of monocytes early after induction of hindlimb ischemia markedly enhances reperfusion following ischemia. It has been demonstrated, in a rabbit model of acute femoral artery ligation, that the collateral artery growth depends directly on the number of circulating monocytes[25]. Moreover, the distinct pro-arteriogenic potential might result from the differential expression of angiogenic factors such as MMP-9[26,54] through the release of extracellular matrix-bound VEGF[55]. In myocardial infarction, a selective depletion of M2 macrophages had a catastrophic prognosis with an increase of frequent cardiac rupture. The tissue repair was entirely rescued by an external supply of M2-like macrophages[56]. These results are in accordance with the role of M1-like and M2-like cells in tissue repair that corresponds to their sequential and complementary functions, in that M1-like macrophages initiate the healing process by stimulating angiogenesis[38,57] while M2-like cells promote stabilization of angiogenesis and tissue maturation by supporting the increase of regenerating fibers diameter[58] during skeletal muscle regeneration in mice.

Thus, monocyte/macrophage could be used as biomarkers to evaluate and manage the irradiated tissue to offer personalized regenerative medicine treatment to the patient. Infiltration of M1-/M2-like macrophages could be investigated by three different methods. The most suitable for the evaluation of macrophage infiltration is flow cytometry with antibody mixtures, but this approach depends on the amount of harvested tissue. Thus, PCR and immunohistochemical staining might be used to analyze expression of cytokines or inflammatory cells.

In summary, we showed that inflammatory cells play a major role after hindlimb irradiation to allow the neovascularization process and maintain blood vessels to limit cachexia development.

## Methods

**Animals, irradiation**. All animal procedures performed were approved by the institutional animal experimentation and ethics committee (C2EA) at the Institute of Radiation Protection and Nuclear Safety (IRSN). Experiments were conducted according to the French veterinary guidelines and those formulated by the European Community for experimental animal use (APAFIS#12903-20180104180861 32 v1 and APAFIS #19137-2019021313569870 v1). Male C57Bl/6 mice (8 weeks old) were used as experimental animals (Janvier, France). Eight-week-old male CCR2[−/−], CR3CR1[−/−] and their wild-type (WT) C57BL/6 littermates were purchased from the Jackson Laboratories. All animals were kept under stable microenvironment conditions (22 ± 1 °C), with alternating 12 h light and dark cycles and received standard laboratory food and water. Mice were anesthetized with isoflurane inhalation (2%) and their right hindlimb was exposed to a single dose of 80 Gy irradiation using 10-MV X-rays at 2.6 Gy/min on an irradiation field of 2*24 cm (Elekta Synergy Platform, Elekta SAS, Boulogne-Billancourt, France).

**Scoring injury**. The kinetics of the evolution of the lesion was assessed by a semi-quantitative analysis. A score between 0 (normal) and 1 (max injury) was assigned to each clinical sign, including wound extent, ulceration, moist desquamation and limb retraction resulting in a total injury scoring up to 5 during the wound development and healing process each week post irradiation.

**Immunohistochemistry**. Gastrocnemius were excised, rinsed in PBS, and frozen in liquid nitrogen. Tissues were cut using a cryostat into 7-μm-thick sections.

Macrophages and CD3-positive cells were visualized after CD68 antibody (1/250, Biorad) or CD3 staining (1/250, DAKO) followed by staining with a donkey 488-AF−anti-rat or anti-rabbit IgG (1/250, Jackson ImmunoResearch) respectively. CD68- and CD3-positive cells were counted in randomly chosen fields with the use of Image J software (NIH). Sections were stained with hematoxylin and eosin (H&E) to analyze muscle structure.

**Analysis of neovascularization**. After irradiation, neovascularisation was evaluated by two different methods as previously[59].

*Microangiography*. At the time of sacrifice, mice were anesthetized (Alfaxan (80 mg/kg body weight) and Xylazine (10 mg/kg body weight), and longitudinal laparotomy was performed to introduce a polyethylene catheter into the abdominal

aorta and inject contrast medium (barium sulfate, 1 g/mL). Angiography of hindlimbs was then performed, and images (two per animal) were acquired with the use of a high-definition digital X-ray transducer. Images were assembled to obtain a complete view of the hindlimbs. The number of pixels occupied by vessels was measured in the quantification area with the use of Primed angio software (Trophy System, Paris, France). The area of quantification was limited by the placement of the iliac, the knee, the edge of the femur, and the external limit of the leg. The results were then expressed as a ratio of irradiated to nonirradiated leg.

*Capillary density measurement.* Gastrocnemius muscle was excised from the right (irradiated limb) and the left (NIR limb), and frozen in O.C.T. compound (Sakura Finetek France, Villeneuve d'ASCQ, France) for cryosectioning. Frozen muscle sections (7 μm) were stained using a DyLight 594 Labeled GSLI-isolectin $B_4$ (Vector Laboratories, dilution 1:200) to identify capillaries. The number of capillaries per $mm^2$ in muscle area was determined.

**Flow cytometry.** Mice were euthanized by cervical dislocation after sedation with isoflurane inhalation (4%) on days 1, 3, 5, 7, 14, 30, 60, 90, 120, 150, or 180 after hindlimb irradiation ($n = 8$–10 mice per time point). Peripheral blood was drawn via inferior vena cava puncture with heparin solution. Whole blood was lysed after immunofluorescence staining using the BD FACS lysing solution (BD Biosciences), and total blood leukocyte numbers were determined using Kova slides. Bone marrow cells were drawn from the femur and filtered through a 40-μm nylon mesh (BD Biosciences). Spleens were collected, and gently passed through a 40-μm nylon mesh (BD Biosciences). For both splenocytes and bone marrow-derived cells, the cell suspension was centrifuged at 400×g for 10 min at 4 °C. Red blood cells were lysed using red blood cell lysing buffer (Sigma-Aldrich) and splenocytes and bone marrow cells were washed with PBS. Nonirradiated and irradiated muscles were collected, minced with fine scissors minced with fine scissors, and gently passed through the Bel-Art Scienceware 12-well tissue disaggregator (ThermoFisher Scientific). Cells were then filtered through a nylon mesh (40 μm) and centrifuged (10 min, 400×g, 4 °C).

Cells isolated from the tissue of interest were incubated in the dark at 4 °C for 30 min with the following antibody mix. For detection of inflammatory cells, total cells were gated on AF700-conjugated anti-CD45, FITC-conjugated anti-CD11b (ThermoFisher Scientific), APC-conjugated anti-Ly-6B.2 (clone 7/4, Biorad), PE-conjugated anti-Ly6G, PB-conjugated anti-CD64, APC-conjugated anti-F4/80, PerCP5.5-conjugated anti-MHCII (BD Biosciences), FITC-conjugated anti-CD4 (clone RM4-5; eBioscience), PercpCy5-conjugated anti-CD8 (clone 53-6.7, BD Pharmingen), APC-conjugated anti-CD3 (clone 17A2, eBioscience), PE-Cyanine7-conjugated anti-CD11c (cloneN418, eBioscience) during 30 min at 4 °C.

We used different antibody mixes in different tissues as follows:

A. Bone marrow: mix 1: Mo: FITC-conjugated anti-CD11b, APC-conjugated anti-Ly-6B.2, PE-conjugated anti-Ly6G; mix 2: lymphocytes: AF700-conjugated anti-CD45, FITC-conjugated anti-CD4, PercpCy5-conjugated anti-CD8, APC-conjugated anti-CD3,

B. Spleen: mix 1: Mo: FITC-conjugated anti-CD11b, APC-conjugated anti-Ly-6B.2, PE-conjugated anti-Ly6G; mix 2: lymphocytes: AF700-conjugated anti-CD45, FITC-conjugated anti-CD4, PercpCy5-conjugated anti-CD8, APC-conjugated anti-CD3,

C. Blood: mix 1: Mo: FITC-conjugated anti-CD11b, APC-conjugated anti-Ly-6B.2, PE-conjugated anti-Ly6G; mix 2: lymphocytes: AF700-conjugated anti-CD45, FITC-conjugated anti-CD4, PercpCy5-conjugated anti-CD8, APC-conjugated anti-CD3,

D. Muscle: mix 1: Mo: AF700-conjugated anti-CD45, FITC-conjugated anti-CD11b, APC-conjugated anti-Ly-6B.2, PE-conjugated anti-Ly6G; mix 2: lymphocytes: AF700-conjugated anti-CD45, FITC-conjugated anti-CD4, PercpCy5-conjugated anti-CD8, APC-conjugated anti-CD3, mix 3: Macrophages: AF700-conjugated anti-CD45, FITC-conjugated anti-CD11b, PE-conjugated anti-Ly6G, PB-conjugated anti-CD64, APC-conjugated anti-F4/80, PerCP5.5-conjugated anti-MHCII, PE-Cyanine7-conjugated anti-CD11c Mo$^{hi}$ (CD11b$^+$Ly6G$^-$7/4$^{hi}$), Mo$^{lo}$ (CD11b$^+$Ly6G$^-$7/4$^{lo}$), M1-like cells (CD11b$^+$Ly6G$^-$F4/80$^+$CD64$^+$MHCII$^+$) and M2-like cells (CD11b$^+$Ly6G$^-$F4/80$^+$CD64$^+$MHCII$^-$), DC (CD11c$^+$MHCII$^+$), CD4 (CD3$^+$CD4$^+$) and CD8 (CD3$^+$CD4$^+$).

The total number of cells was then normalized to muscle weight. Cells were analyzed using a flow cytometer (Canto II, BD Biosciences).

**Real-time quantitative polymerase chain reaction.** Total RNA was extracted from frozen muscle samples with Trizol reagent according to the manufacturer's instructions (Invitrogen, France). After quantification on a NanoDrop ND-1000 apparatus (NanoDrop Technologies, Rockland, DE), reverse transcription was performed using the QuantiTect Reverse Transcription (Qiagen) according to the manufacturer's instructions. PCR was performed with the ABI PRISM 7900 using Power SYBR PCR Master Mix (Bioline). Ct for mouse GAPDH was used to normalize samples amplification. The following oligonucleotides (Applied Biosystems) served as primers: *GAPDH* forward: 5′-CGTCCCGTAGACAAAATGGTGAA-3′, reverse: 5′-GCCGTGAGTGGAGTCATACTGGAACA-3′; *eNOS* forward: 5′-CGCCCACCCAGGAGAGATCCAC-3′, reverse 5′-

GCATCGGCAGCCAAACACCAAAGT-3′; *CCL-2* forward: 5′-CCCCACTCACCTGCTGCTA-3′, reverse 5′-TTACGGGTCAACTTCACATTCAAA-3′; *CX3CL1* forward: 5′-GTGGCTTTGCTCATCCGCTATCAG-3′, reverse: 5′-CACATTGTCCACCCGCTTCTCA-3′.

**Western blot.** Preparation of protein extracts was performed. Briefly, to prepare total protein, *tibialis anterior* muscles from irradiated and nonirradiated hindlimbs were homogenized in RIPA buffer (50 mM Tris HCl pH7.4, 150 mM NaCl, 1 mM EDTA, 1% Triton X-100, 1% deoxycholate, 0.1% SDS with protease and phosphatase inhibitors). Proteins were resolved in gradient denaturing gel electrophoresis and blotted onto 0.2μm nitrocellulose sheets (Biorad, Marnes la Coquette, France). Antibodies against *eNOS* (1:1000; BD Biosciences) were used for immunoblotting. As a protein loading control, membranes were stripped and incubated with a monoclonal antibody directed against *GAPDH* (1:10,000, Abcam) and specific chemiluminescent signal was detected.

**Monocyte adoptive transfer experiment.** For the adoptive transfer experiment, spleens from 8-week-old C57BL/6 mice were mechanically dissociated on a 40-mm cell strainer. Splenocytes stained with AF700-conjugated anti-CD45, FITC-conjugated anti-CD11b, APC-conjugated anti-Ly-6B.2, Pacific Blue-conjugated anti-Ly6G and PE-conjugated anti-NK1.1. CD11b$^+$Ly6G$^-$NK1.1$^-$ 7/4hi and CD11b + Ly6G2NK1.1$^-$ 7/4lo were then selected using a FACS BD InFlux. Purity for both subset was 99%. Before injection, cells were counted by trypan blue exclusion. In total, $10^5$ Mo$^{hi}$ monocytes or $10^5$ Mo$^{lo}$ monocytes were then intramuscular injected to C57BL/6 recipients the next day after hindlimb irradiation.

**Statistic and reproducibility.** Results were expressed as ± SEM. One-way analysis of variance (ANOVA) was used to compare each parameter. Post hoc Bonferroni's *t* test comparisons were then performed to identify which group differences account for the significant overall ANOVA. A value of $P < 0.05$ was considered significant.

The investigation of the statistical association between infiltration of inflammatory cells and leukocyte subsets in one hand with postischemic neovascularization in other hand is challenging since these two measures were performed on different animal cohorts. This makes the classical statistical approaches inefficient since predictive and response variables are not observed on the same subject. The Bayesian latent variable model[60] offers the possibility to include priors on the distribution of the inflammatory cells and leukocyte subset counts to estimate their correlation with the vessel growth. The model fitting was performed using JAGS software via Markov chain Monte Carlos. All *P* values in the different statistical analysis sections were corrected for multiple testing using Benjamini and Hochberg procedure.

**Reporting summary.** Further information on research design is available in the Nature Portfolio Reporting Summary linked to this article.

## Data availability
All data that supported the findings of this study are included within this paper and its Supplementary Information files and Supplementary Data 1.

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

## Acknowledgements

Dr Celine Loinard is supported by Institut de Radioprotection et de Sûreté Nucléaire (IRSN), Fontenay-aux-Roses, France.

## Author contributions

C.L and R.T. conceived the study, designed the experiments, and interpreted the data. C.L., M.A.B., B.L., S.F., and J.B. performed the experiments and analyzed the data. C.L. and R.T. wrote the manuscript with final approval from all co-authors.

## Competing interests

The authors declare no competing interests.

**Additional information**

