## [Peer Review File · Communications Biology]

Reviewers' comments:

Reviewer #1 (Remarks to the Author):

This report has analyzed the dynamics of inflammatory cells and neovascularization in radiation-induced muscle injury in mice. However, it is overall too descriptive and is recommended for rejection.

This manuscript needs to be edited in English.

Overall, the use of paragraphs should be modified according to the journal's requirements.

The importance of the radiation-induced muscle injury model should be described in more detail in the first paragraph of Introduction.

Lane 7 in page 4 mentions the ischemia model, but should describe in detail what it has in common with the radiation model.

The authors should discuss how the results of this study will ultimately be applied to humans.

In the first paragraph of page 6, the authors should describe the details of the scoring criteria.

In line 15 of page 6, "leucocytes" \diamond "leukocytes".

The reader cannot understand the meaning of the sentence, "After wards, monocytes differentiated ..." in line 17-18 of page 6.

What is "NIR"? Abbreviations must be explained.

References are required after the sentence on page 7, line 21.

Reference 28 is a reference to Cx3cl1, so a reference to MCP-1 is needed on page 8, line 12.

The sentence on page 8, line 22-23 should be corrected so as not to mislead the readers. "adhesion molecules" \diamond "chemokines such as ccl2 and cx3cl1"

The lack of any histopathology of normal and post-injured muscles over time makes it difficult to understand the inflammatory response and the healing process of the radiation-induced muscle injury.

Why did the authors choose the 180-day long period? Is there any effect of infection on the injured site during the course of the long-term experiment? The various events in the damaged area in this model peak around 20-30 days and seem to have largely converged around 50-60 days (DOI: 10.3389/fonc.2021.651637; DOI: 10.1080/09553002.2018.1420925). It is questionable whether it is reasonable to define D0-30 as the acute phase and D60-180 as the chronic phase. It may be reasonable to define the acute phase as up to D7 and the remodeling phase as up to about D7-30. In addition, the meaning of the bimodal infiltration of DCs and lymphocytes into muscle (graphical abstract) should be examined and clarified.

Overall, there appears to be inconsistency and discrepancy in the various longitudinal data. For example, cx3cl1 mRNA peaks at D90, but lymphocyte infiltration peaks at D7 and is baseline at D90. Furthermore, the peak of eNOS protein expression is D30, while the peak of angiogenesis is D14.

Since the authors focus on angiogenesis, intramuscular angiogenesis should be shown histologically (e.g. immunostaining with anti-cd31) in addition to microangiography to facilitate the reader's understanding.

The authors should describe the details of the angiographic scoring criteria (Fig. 5E).

VEGF mRNA dynamics (Fig. 5B) do not match protein dynamics (Fig. 5D). Furthermore, the Vegf mRNA peaks at D90 (Fig. 5B), but the capillary density is significantly suppressed (Fig. 5F), a contradiction. The authors must explain these contradictory data. Also, the Vegf protein cannot be considered biologically meaningful if there is no significant increase or decrease (Fig. 5D).

Fig. 5F, "cappillary" \diamond "capillary".

In D3, the number of Mohi and Molo in the blood of *ccr2*^{-/-} mice, but not *cx3cr1*^{-/-} mice, was significantly lower than in wild-type mice (Fig. 6A), what was found and considered? The authors should also indicate whether there were changes in leukocyte infiltration into the muscle with *ccr2*^{-/-} and *cx3cr1*^{-/-} mice.

What about BM-derived EPC infiltration, which is important for angiogenesis (DOI: 10.1172/JCI43027), into the injured muscle throughout the experimental period?

The scRNAseq experiment will address some of the questions that reviewers and readers may have.

Regarding the third paragraph on page 9, I wonder if a Bayesian approach is appropriate for this study. Why not actually conduct a mouse experiment and use various M1 and M2 markers to prove it? Furthermore, the fourth paragraph is too descriptive.

"In addition, Molo numbers in *cx3cr1*^{-/-} mice were ..." in page 10, line 18-19. Since no significant decrease was observed, this sentence should be omitted to avoid misleading the reader.

Reviewer #3 (Remarks to the Author):

The authors investigated the role of inflammatory cell populations in response to radiation-induced injury of the muscle at early and late phases post-irradiation. They identify the dynamic nature of the immune response and recruitment of key responders to the site of injury and the role they play in both early neovascularization and injury progression establishing a relationship between specific immune cell populations and vascularization states. The presented work is very well done and is of great interest and relevance to the radiation therapy field.

There are some technical points, however, that should be addressed as well as some questions regarding some of the results:

1. There are discrepancies between figures referred to in the text and the actual figures themselves. In particular, referring to statistical significance in the text but lacking significance markings in the graphs. Also, they sometimes refer to incorrect figure labels within the text.
 - a. Figure 2D is listed as 1D in text- in the text it indicates Mo hi, Mo lo and Neutrophils peaked at D1, D5 and D60 however in Figure 2 there is only statistical significance at D1
 - b. Figure 4C (T lymphocytes)- in the text they state The number of CD3 positive lymphocytes increased by 39-fold at D7, 24-fold at D14 and 18-fold at D21 however in the graph there is no D21, only D30
 - c. Figure 6E is referred to as 6D in text
2. There are acronyms such as NIR that have not been defined before their use and the manuscript
3. Material and Methods- Irradiation: The author should specify the size of the irradiation field and/or the portion of mice body surface irradiated.
4. Material and Methods- Mice are all male. The author should justify their specific choice and discuss whether they are expecting different results in the case of employing female mice.

5. Material and Methods- Flow cytometry mix of Abs: The author are talking about one mix but they should divide the Abs in groups according to the different combinations used in the different panels employed.

Regarding results/discussion:

6. NIR controls are referred as littermates of the irradiated mice. However, there are studies showing the existence of bystander effects from irradiated animals to non-irradiated animals when they are cage mates, especially after high dose RT. The author should address this point in the discussion.

7. How do the authors explain such an early influx of M-2 like MFs in the muscle? In the discussion they mention the importance of this pro-angiogenic MFs subpopulation in the resolution of Myocardial infarction (MI), however in MI but also in other types of sterile injuries the content of M2-like MFs peaks at 5/7 days post-injury not earlier. Also, because they are gradually differentiating from M1-like Mo/MFs which are the first ones reaching the injured tissue. Author should discuss their findings in relation to this and hypothesize about the reason of the difference between them and what is known in the literature.

8. Furthermore, I would like them to comment on the sustained levels of neutrophils, Mo(hi) and Mo(lo) in the blood (depicted in Figure 2C) and provide a hypothesis or details on why the fluctuations in these cells observed in the spleen and muscle are not reflected in the blood.

9. Moreover, it would have been great to see how transfer of M2-like MFs can help the healing process after such a high radiation dose. In case the data could not be provided, the author should discuss expected outcomes after such a transfer.

Dear editor and reviewers,

Please find below our point-by-point response to the reviewers' comments.

Modifications in the main text are highlighted in yellow

Reviewer #1 (Remarks to the Author):

This report has analyzed the dynamics of inflammatory cells and neovascularization in radiation-induced muscle injury in mice. However, it is overall too descriptive and is recommended for rejection.

1/ This manuscript needs to be edited in English.

The manuscript has been edited by a native English speaker.

2/ Overall, the use of paragraphs should be modified according to the journal's requirements.

As requested, the paragraphs have been modified according to the journal's requirements

3/ The importance of the radiation-induced muscle injury model should be described in more detail in the first paragraph of Introduction.

Local overexposure to ionizing radiation has severe health consequences, especially when the absorbed dose exceeds 25 Gy and leads to tissue necrosis^{1,2}.

Clinical signs of a patient exposed to ionizing radiation have been already described and it has been mentioned extensive necrosis in the right limb, intense pain, multiple infections and ankyloses of the right hip and knee¹. Clinically, the symptoms appear days after the exposure and the lesion is described as dynamic, with a progressive extension of the radio-induced damage associated with ischemic, inflammatory and necrotic processes³⁻⁵. At the molecular level, nuclear DNA damage, oxidative stress caused by reactive oxygen and nitrogen species and mitochondrial dysfunction with perturbations in oxidative metabolism have been previously described^{6,7}. In the particular case of radiation muscular injury, the evolution of the damage in patients has been associated to a decrease of muscular function⁸.

As requested, we have clarified the importance of the radiation-induced muscle injury model and with the changes highlighted in yellow in the manuscript: page 3, line 16-21

In experimental model, it has been shown that the numbers of both myonuclei and satellite cells per myofiber were decreased in a dose-dependent manner⁹. Moreover, a single irradiation dose of 18 Gy block muscle regeneration by promoting lethality of myoblasts¹⁰. Studies of muscular pathology using

more than 25 Gy irradiation dose, have shown morphological alterations, hemorrhage, necrosis, inflammation, fibrosis and mitochondria destruction¹¹⁻¹⁴. In our preclinical model of 120 Gy local radiation hindlimb in nude mice (unpublished data) we demonstrated a characteristic muscular pathological feature including severe atrophy, central nuclei, fibrosis and inflammatory infiltrates. Furthermore, an acute damage to the microvascular network of the irradiated muscle was evident with a decrease in the endothelial cells number and an abnormal cellular morphology. These findings were in accordance to the results obtained in a victim from accidental exposure to high dose ionizing radiation. Additionally, the irradiated mouse muscle presented a fiber-typing switch with an increase in the type IIA and IIX fast fibers compared to controls. Some embryonic MHC positive fibers. After, high dose exposure to radiation, the tissue maintains a deleterious T cells pro-inflammatory response during the long term follow up until 6 months.

4/ Lane 7 in page 4 mentions the ischemia model but should describe in detail what it has in common with the radiation model.

As requested, we describe in detail the similarities between ischemia disease and radio-induced local injury (RLI) and the changes highlighted in yellow in the manuscript: page 3-4, line 17-2. Ischemia is the common process in both diseases RLI and cardiovascular disease. In ischemic disease, insufficient organ perfusion following thrombotic vessel obstruction of the feeding artery is a major determinant of post-ischemic remodeling¹⁵. However, exposure of mammalian cells such as endothelial cells to ionizing radiation leads primarily to DNA damage-induced cell death¹⁶. Ischemia is characterized by vascular damage/rarefaction and inflammation resulting in fibrosis characterized by collagen-based scar¹⁷. The ischemic tissue response is based on four principal processes, vasculogenesis, angiogenesis, arteriogenesis, and collateral growth, which contribute to tissue repair and remodeling during acute and chronic ischemic vascular diseases¹⁸. These processes result from hemodynamical forces changes within the vascular wall leading to modification of the vascular homeostasis^{19 18}.

As requested, we added this sentence page 5, line 7-8: Therapeutic modulation of the inflammatory response may hold promise to improve reparative response for the prevention of postischemic disease.

5/ The authors should discuss how the results of this study will ultimately be applied to humans.

The aim of this study is to understand the chronologic events in early and late time post-IR and we focus on the role of Mo/ M ϕ . We showed that ionizing radiation induces an early phase characterized by a pro-inflammatory recruitment and infiltration of CD4-, CD8-T cells, Mo^{hi} and M1-like leading to the neovascularization process to prevent tissue hypoxia. Afterwards, we observed a late phase

supported by a positive correlation between the decreased recruitment of anti-inflammatory Mo^{lo} and differentiation into M2-like, vascular rarefaction and cachexia characterized by sustained pro-inflammatory response. In this context, we would like to propose a personalized medicine and possibility to use Mo/M ϕ as biomarker to i) **predict tissue complication** after radiation exposure and/or to ii) **determine the therapeutic strategy**.

In the discussion of the manuscript, we suggest that Mo/M ϕ could be used **as biomarkers to predict the radiation induced tissue complication**. In fact, Mo/M ϕ have been already published as a good biomarker in a clinical study, conducted on a cohort of 49 patients included in breast cancer radiotherapy protocol, which showed that the increase of mRNA RG1 level expression in M ϕ 24 hours after a first session of radiotherapy, was correlated with the risk of developing severe erythema 8 weeks after the end of radiotherapy²⁰. Finally, in patients with radiation-induced mucositis, the number of M ϕ has been associated with the severity of the disease²¹.

Moreover, we propose to optimize the medical management by a **personalized regenerative treatment** based on the modulation or reprogramming of the anti-inflammatory/pro-inflammatory phenotype of Mo/M ϕ . Several studies have shown that nanoparticles injection loaded immunomodulatory drugs lead to reprogramming the anti-inflammatory/pro-inflammatory function of Mo/M ϕ to repair tissue or for cancer treatment²²⁻²⁵. For example, Jiang et al. ²⁶, in CCl₄-induced mouse liver injury model, showed that nanoparticles loaded with tripolyphosphate dynamically regulated M1 into M2 macrophage reprogramming and relieved the liver injury. Finally, in a colon cancer mouse model, M2 macrophages treated with PEGylated liposomes containing IFN- γ expressed elevated NO and decreased arginase levels, promoted M2 to M1 polarization and significant antitumor responses²⁷. In our condition, we would suggest nanoparticles injection, that will potentially carry anti-inflammatory or pro-inflammatory function (depending on the inflammatory state of the patient after blood collection or tissue biopsy) with the aim to modulate the inflammatory reaction and stimulate the regenerative tissue process.

6/ In the first paragraph of page 6, the authors should describe the details of the scoring criteria.

As requested, we give more details for the scoring and we added the changes highlighted in yellow in the manuscript in materiel and methods page 17, line 17-20. A score between 0 (normal) and 1 (max injury) was assigned to each parameter regarding wound extent, ulceration, moist desquamation and limb retraction (see Figure below), resulting in a total injury scoring during the wound development and healing process each week post-irradiation.

7/ In line 15 of page 6, “leucocytes” \diamond “leukocytes”.

We modified the word.

8/ The reader cannot understand the meaning of the sentence, “After wards, monocytes differentiated ...” in line 17-18 of page 6.

As requested, we changed the meaning of the sentence by: we did not observe monocyte accumulation in muscle, suggesting they differentiated in macrophages in line 17-18 of page 6 and highlighted in yellow in the manuscript

9/ What is “NIR”? Abbreviations must be explained.

We added the complete name of Non-irradiated (NIR) in the manuscript page 6 line 18

10/ References are required after the sentence on page 7, line 21.

As requested, line 21, page 7 a reference has been added and highlighted in yellow in the manuscript.
Loinard C, Vilar J, Milliat F, Levy B, Benderitter M. Monocytes/macrophages mobilization orchestrate neovascularization after localized colorectal irradiation. *Radiation research*. 2017;187:549-561

11/ Reference 28 is a reference to Cx3cl1, so a reference to MCP-1 is needed on page 8, line 12.

As requested, line 12, page 8 a reference has been added and highlighted in yellow in the manuscript

Cochain C, Rodero MP, Vilar J, Recalde A, Richart AL, Loinard C, Zouggari Y, Guerin C, Duriez M, Combadiere B, Poupel L, Levy BI, Mallat Z, Combadiere C, Silvestre JS. Regulation of monocyte subset systemic levels by distinct chemokine receptors controls post-ischaemic neovascularization. *Cardiovascular research*. 2010;88:186-195

12/ The sentence on page 8, line 22-23 should be corrected so as not to mislead the readers. “adhesion molecules” ◊ “chemokines such as ccl2 and cx3cl1”

As requested, page 8, line 22-23, the sentence has been corrected and highlighted in yellow in the manuscript

13/ The lack of any histopathology of normal and post-injured muscles over time makes it difficult to understand the inflammatory response and the healing process of the radiation-induced muscle injury.

As requested, to better understand the kinetics of inflammatory response and the healing process of the radiation-induced muscle injury, we performed specific staining with visualization of macrophages and T cells after using a CD68 or CD3 antibody respectively. CD68– and CD3–positive cells were counted in randomly chosen fields with the use of Image J software (NIH). Additionally, the sections were stained with hematoxylin and eosin (H&E) to analyze muscle structure.

The hematoxylin-eosin staining and CD68 and CD3 – positive cells were performed in Non-IRradiated (NIR) and irradiated tissue at D1 to D180 post-irradiation (we added in supplementary Figure 1)

14/ Why did the authors choose the 180-day long period? Is there any effect of infection on the injured site during the course of the long-term experiment? The various events in the damaged area in this model peak around 20-30 days and seem to have largely converged around 50-60 days (DOI: 10.3389/fonc.2021.651637; DOI: 10.1080/09553002.2018.1420925). It is questionable whether it is reasonable to define D0-30 as the acute phase and D60-180 as the chronic phase. It may be reasonable to define the acute phase as up to D7 and the remodeling phase as up to about D7-30. In addition, the meaning of the bimodal infiltration of DCs and lymphocytes into muscle (graphical abstract) should be examined and clarified.

A/ Why did the authors choose the 180-day long period?

During many months to years, clinical signs are characterized by intense pain, multiple infections and ankyloses of the right hip and knee¹, successive and unpredictable inflammatory waves over the first few days to years after irradiation, and these lead to horizontal and vertical extension of tissue injury including vascular rarefaction and muscle cachexia².

Experimental data: with the clinical hindlimb radio-induced injury, the mouse model was used in order to mimic the clinical features with inflammatory waves, vascular rarefaction, cachexia injury and healing aspects up to 6 months post-irradiation as observed in the patient. We showed previously on unpublished results from the lab, on Nude immunodeficient mice, an increase in the injury score two weeks post-irradiation (40 or 60 Gy), with a return to a non-irradiated situation maintained for up to 6 months after irradiation, compared to BalB/c (immunocompetent).

Moreover, in immunocompetent mice a second wave of injury score was observed at D110 post-irradiation (see figure below). These results underline the role of T cells in the second radiation-induced inflammatory wave and suggest a preponderant role of T cells in tissue damage in this context.

B/ Is there any effect of infection on the injured site during the course of the long-term experiment?

No infection was observed, otherwise the inflammation would have remained at a high level during all the follow up until the 6 months after exposure.

C/ The various events in the damaged area in this model peak around 20-30 days and seem to have largely converged around 50-60 days (DOI: 10.3389/fonc.2021.651637; DOI: 10.1080/09553002.2018.1420925). It is questionable whether it is reasonable to define D0-30 as the acute phase and D60-180 as the chronic phase. It may be reasonable to define the acute phase as up to D7 and the remodeling phase as up to about D7-30.

In our study, we define the acute and late phases in function of the vascular process. Therefore, we determined D0-30 as the early phase corresponding to the neovascularization process and pro-inflammatory cells recruitment, and D60-180 as the late phase characterized by vascular rarefaction correlated to recurrent and chronic pro-inflammatory phenotype. Irradiation disease are really uncommon regarding inflammatory waves compared to other diseases such as myocardial infarct or peripheral occlusion disease. In cardiovascular ischemic disease, during the first acute phase (hours-4 days), it has been shown a pro-inflammatory recruitment, vascular regeneration and, subsequently a late phase (5–7 days after the ischemic event) corresponding to a switch and anti-inflammatory

recruitment and tissue regeneration^{28 29 30}. In these diseases the acute and late phase are determined in function of the inflammatory switch pro- vs anti-inflammatory process. Moreover, the main cells infiltrated in the tissue, after MI, corresponded to neutrophils, Mo at D3 post-MI compared to T cells infiltration and returned to basal level at D7 or D14 post-MI¹⁷. After ionizing radiation, we do not observe the remodeling phase characterized by the switch from pro-inflammatory to anti-inflammatory cell phenotype, but the maintenance of the pro-inflammatory response during a long time period after exposure. That is the reason why we decided to focus on the vascular process to establish the early and late phases. Moreover, in irradiated muscle the main cells infiltrated is corresponding to T cells and stay up until the end of experiment.

In the papers cited by the reviewer, Raznitsyna et al. after 25 Gy exposure, they showed a first inflammatory peak of the white blood cells at D7 post-IR, that returned to basal level at D30. However, at D45 post-IR the number of white blood cells increased again, suggesting a dynamic recurrent pro-inflammatory process. These results confirm our findings in our preclinical studies on the dynamic inflammatory waves. However, it will be interesting to extend the follow up to identify the inflammatory response specific to the irradiation condition.

Moreover, the reviewer mentioned the study by Wang et al, in which they defined, after 30 Gy exposure, early and late phases based on scoring injury. They showed a correlation between scoring and inflammatory cytokines expression in the muscle at D7. As we explained previously, it would be interesting to analyze at long term the recurrency of the inflammatory waves by the expression of pro- and anti-inflammatory cytokines at long term.

Finally, in our study, it is the first time that an evaluation of the inflammation and vascularization process was performed **on a very long time point (until 6 months)** after radiation exposure. The objective of our study is clearly to mimic the clinical situation with long term effects observed after radiation therapy or accidental situation in industrial activities during months and years after the exposure.

This study highlights the pro-inflammatory waves after exposure and their effect on the vascular process at long term. That is the reason why we decided to define the acute and late phases, by taking into consideration the timeline of this study and the associated pathophysiological process, too.

D/ In addition, the meaning of the bimodal infiltration of DCs and lymphocytes into muscle (graphical abstract) should be examined and clarified.

The bimodal infiltration of DCs and lymphocytes into muscle is a consequence of radiation exposure. In the study described in reference 1, such successive inflammatory waves were observed on the patient over the first few days, weeks to years after irradiation. In this study, the preclinical model developed tries to mimic the clinical situation (characterized by chronic inflammatory waves, vascular

rarefaction, and cachexia). Based on the graphical abstract, we can observe that the level of LT and DC peaks at different time points and never return to a non-irradiated condition, which explains the state of a proinflammatory tissue and vascular damage with no regeneration process. This is the specificity of the tissue response in irradiation condition.

The difference with cardiovascular ischemic diseases (hindlimb or MI), is that the inflammatory cells peaked at D3 and then return to basal level from 7 to 14^{17 28}.

15/ Overall, there appears to be inconsistency and discrepancy in the various longitudinal data. For example, cx3cl1 mRNA peaks at D90, but lymphocyte infiltration peaks at D7 and is baseline at D90. Furthermore, the peak of eNOS protein expression is D30, while the peak of angiogenesis is D14.

A/ Overall, there appears to be inconsistency and discrepancy in the various longitudinal data. For example, cx3cl1 mRNA peaks at D90, but lymphocyte infiltration peaks at D7 and is baseline at D90.

According to the kinetic of evolution of CX3CL1, we observed

- The tendency to increase of mRNA expression level of CX3CL1 associated with a significant increase of CD3 positive cells in the muscle from D7 to D30 post-IR **suggesting the recruitment of T cells from the blood into the muscle.**
- The increase of T cells **in the blood** from D7 to D30 (cytometry estimation Figure 3B) associated with an increase of T cells in the muscle (cytometry estimation Figure 3C) and confirmed by T cells staining in the muscle (figure 4C) **suggesting the recruitment of T cells from the blood into the muscle.**

Concerning the peaks observed at D90 with no effect on the T Cells infiltration in the muscle. We suggest several hypotheses:

- during the late phase, we observed a production of T cells in the spleen (Figure 3A) and suggest a blocking of their egress from the spleen to the blood (as no significant increase of these in the blood). (Figure 3A, 3B)
- the lack of presence of T cells in the blood could explain the absence of the recruitment in the muscle whatever the peak of CX3CL1 at D90 post-IR.

B/ Furthermore, the peak of eNOS protein expression is D30, while the peak of angiogenesis is D14.

The angiogenic process results of multiple pro-angiogenic factors and in this study, we analyzed eNOS and VEGF mRNA and protein known to be some of the key players in neovascularization process to repair ischemic tissue. Regarding the kinetic of evolution of these two actors, our results do not allow us to explain the stimulation of the vascularization process at the different time points corresponding to the protein analyses. We agree that a more accurate evaluation of the eNOS protein (additive time

points) could be more informative and support our hypothesis on the involvement of eNOS in the angiogenic process as described in several papers. eNOS has been already published to play a major role in healing process³¹. Additionally we could suggest the involvement of other factors such as, VEGF-B³², b-FGF^{33,34} or Ang³⁵ largely described in the literature, for their major role in the stimulation of angiogenesis (we did not investigate in this study).

16/ Since the authors focus on angiogenesis, intramuscular angiogenesis should be shown histologically (e.g., immunostaining with anti-cd31) in addition to microangiography to facilitate the reader's understanding.

We totally agree with the reviewer on the importance to evaluate intramuscular angiogenesis. Different studies assessed angiogenesis using lectin as an immunofluorescent tool for marking endothelial cells^{36 37 38}. That is the reason why, in this study, GSLI-isolectin B₄ (Vector Laboratories) has been used, as described in materials and methods, to identify the number of capillaries per mm² in muscle area (Figure 5F).

17/ The authors should describe the details of the angiographic scoring criteria (Figure 5E).

As requested, we give more details for the angiographic score and we added the changes highlighted in yellow in the manuscript in materials and methods, page 17, line 9-18. Mice were anesthetized (Alfaxan, 80 mg/kg body weight) and Xylazine (10 mg/kg body weight), and longitudinal laparotomy was performed to introduce a polyethylene catheter into the abdominal aorta and inject contrast medium (barium sulfate, 1 g/mL). Angiography of hindlimbs was then performed, and images (2 per animal) were acquired with the use of a high-definition digital x-ray transducer. Images were assembled to obtain a complete view of the hindlimbs. The number of pixels occupied by vessels was measured in the quantification area with the use of Primed angio software (Trophy System, Paris, France). Area of quantification was limited by placement of the iliac, the knee, the edge of the femur, and the external limit of the leg. The results were then expressed as a ratio of irradiated to nonirradiated leg.

18/ VEGF mRNA dynamics (Fig. 5B) do not match protein dynamics (Fig. 5D). Furthermore, the Vegf mRNA peaks at D90 (Fig. 5B), but the capillary density is significantly suppressed (Fig. 5F), a contradiction. The authors must explain these contradictory data. Also, the Vegf protein cannot be considered biologically meaningful if there is no significant increase or decrease (Fig. 5D).

We agree with the reviewer on the fact that there is no matching of the dynamic between mRNA and protein levels. However as previously explained for eNOS expression, the time points could be more

accurate to observe the correlation. In our study, we demonstrate that VEGF mRNA expression level increase at D90, with no upregulation of VEGF protein level. This can explain the rarefaction of the vascularization. In order to not confuse the scientific message, we removed VEGF results.

19/ Fig. 5F, “cappillary” \diamond “capillary”.

we modified

20/ In D3, the number of Mo^{hi} and Mo^{lo} in the blood of *ccr2*^{-/-} mice, but not *cx3cr1*^{-/-} mice, was significantly lower than in wild-type mice (Fig. 6A), what was found and considered? The authors should also indicate whether there were changes in leukocyte infiltration into the muscle with *ccr2*^{-/-} and *cx3cr1*^{-/-} mice.

The role of CCR2 in the mobilization of Mo^{hi} from the BM to the blood has been largely published^{39 40 41 42 17}. In an experimental model of acute skeletal muscle injury induced by barium chloride injection, it was shown by flow cytometry that^{43,44}:

- at baseline, the percentage of 7/4⁺Ly-6G⁻ Mo/M ϕ was increased in bone marrow but reduced in blood in *Ccr2*^{-/-} mice as compared with wild-type controls

- at d3 postinjury, the percentage of 7/4⁺Ly-6G⁻ Mo/M ϕ was also increased in bone marrow and reduced in blood in *Ccr2*^{-/-} mice

These results highlight the role of CCR2 in the mobilization of Mo^{hi} from the BM to the blood.

To answer to the reviewer, we analyzed the infiltration of macrophages by performing CD68 immunostaining on irradiated and non-irradiated muscle from WT, *CCR2*^{-/-} and *CX3CR1*^{-/-} animals, 10 days post-irradiation. We demonstrate the low number of M ϕ in *CCR2*^{-/-} muscle compared to WT ($p < 0.05$). We confirmed the role of CCR2 in the mobilization of Mo from the BM to the irradiated muscle through the blood. This result has been highlighted in the manuscript in yellow (page 10, line 10-11) and added as panel D in Figure 6 (shown below).

21/ What about BM-derived EPC infiltration, which is important for angiogenesis (DOI: 10.1172/JCI43027), into the injured muscle throughout the experimental period?

In a previous study, we evaluated the vasculogenesis process in a wound healing model in irradiated and non-irradiated condition¹⁹. We showed that, using chimeric GFP mice, there is a mobilization of BM-derived EPC to the site of lesion and their incorporation and differentiation into endothelial cells in the vessel (immunostaining GFP+ CD31+). However, few GFP+ CD31+ cells were identified in irradiated muscle; these results are in adequation with different studies, which have shown that vasculogenesis mechanism may be minor^{45 46}. The reported relative contribution of transplanted cells to the endothelium of growing vessels varies widely, from almost no incorporation to 50%^{47 48 49}.

22/ The scRNAseq experiment will address some of the questions that reviewers and readers may have.

This study focused on physiopathology of RLI. We totally agree with the reviewer about the cellular and molecular aspects and would be necessary to investigate for a future study.

23/ Regarding the third paragraph on page 9, I wonder if a Bayesian approach is appropriate for this study. Why not actually conduct a mouse experiment and use various M1 and M2 markers to prove it? Furthermore, the fourth paragraph is too descriptive.

We totally agree with the reviewer concerning the importance and interest of the approach based on M1 or M2 injection to prove the efficiency. However, we consider the Bayesian mathematic approach with some limitation but the easiest way to correlate Mφ and vascularization.

However, we need to confirm the role of Mφ on the vascularization, for example by nanoparticles injection in order to promote the M1 to M2 phenotype switch during the late phase, with the objective to stabilize the vascularization for tissue regeneration. Moreover, Mφ experimentation typically involves *ex vivo* culture (e.g., from the peritoneum), or the differentiation from bone marrow progenitor cells to form bone marrow-derived macrophages. As discussed, macrophage provenance,

culture conditions (L929 supernatant quality is critical), and inflammatory stimuli will naturally affect cell phenotype, function, and inflammatory status. Cell heterogeneity is a major limitation of culturing primary macrophages and it is therefore desirable to robustly control conditions *in vitro* in order to fully characterize the response of macrophages to specific and quantifiable stimuli⁵⁰. In any case, the further experiments that we would provide for clinical transfer will be based on the nanoparticle administration to modulate the phenotype of the M ϕ ^{27,51}.

C/ Furthermore, the fourth paragraph is too descriptive.

We agree and remove the fourth paragraph

24/ "In addition, Molo numbers in cx3cr1-/- mice were ..." in page 10, line 18-19. Since no significant decrease was observed, this sentence should be omitted to avoid misleading the reader.

The sentence has been removed

Reviewer #3 (Remarks to the Author):

The authors investigated the role of inflammatory cell populations in response to radiation-induced injury of the muscle at early and late phases post-irradiation. They identify the dynamic nature of the immune response and recruitment of key responders to the site of injury and the role they play in both early neovascularization and injury progression establishing a relationship between specific immune cell populations and vascularization states. The presented work is very well done and is of great interest and relevance to the radiation therapy field.

There are some technical points, however, that should be addressed as well as some questions regarding some of the results:

1. There are discrepancies between figures referred to in the text and the actual figures themselves. In particular, referring to statistical significance in the text but lacking significance markings in the graphs. Also, they sometimes refer to incorrect figure labels within the text.

a. Figure 2D is listed as 1D in text-

As suggested, we modified Figure 1D to Figure 2D in the text.

More importantly, in the text (page7, line 6-9) it indicates “Mo hi, Mo lo and Neutrophils peaked at D1, D5 and D60 “, however in Figure 1 we obtained statistical significance only at D1 after IR compared to Non-irradiated (NIR) (using anova/ancova statistical test). That is the reason why no significance was marked in the graph.

As mentioned by the reviewer, we corrected in the text all the figure labels.

b. Figure 4C (T lymphocytes)- in the text they state The number of CD3 positive lymphocytes increased by 39-fold at D7, 24-fold at D14 and 18-fold at D21 however in the graph there is no D21, only D30

As suggested by the reviewer. We modified D21 to D30.

c. Figure 6E is referred to as 6D in text

As suggested, we modified Figure 6E to Figure 6D in the text.

2. There are acronyms such as NIR that have not been defined before their use and the manuscript

As suggested, we define in the manuscript NIR: non-irradiated and highlighted in yellow page 4, line

18

3. Material and Methods- Irradiation: The author should specify the size of the irradiation field and/or the portion of mice body surface irradiated.

The irradiation field is 2*24cm, hindlimb of mice is irradiated without the foot (see illustration below):

The materials and methods have been completed accordingly page 17, line13-14.

4. Material and Methods- Mice are all male. The author should justify their specific choice and discuss whether they are expecting different results in the case of employing female mice.

Estrogen are known to have a protective effect on vascular disease^{52 53 54 55} and also, anti-inflammatory⁵⁶, antioxidant⁵⁷, and mitochondrial protective properties^{58 59}. In this study, we evaluate the neovascularization and inflammatory process in tissue regeneration, that's the reason why we avoid the bias of estrogen effect on both parameters after ionizing radiation by using male mice.

5. Material and Methods- Flow cytometry mix of Abs: The author are talking about one mix but they should divide the Abs in groups according to the different combinations used in the different panels employed.

As requested, we give more details for the different antibody mixes in tissues as follows and we added the changes highlighted in yellow in the manuscript in materials and methods, page 19, line 1-20.

- A- Bone marrow: mix 1: Mo: FITC-conjugated anti-CD11b (ThermoFisher Scientific), APC-conjugated anti-Ly-6B.2 (clone 7/4, Biorad), PE-conjugated anti-Ly6G; mix 2: lymphocytes: AF700-conjugated anti-CD45, FITC-conjugated anti-CD4 (clone RM4-5; eBioscience), PercpCy5-conjugated anti-CD8 (clone 53-6.7, BD Pharmingen), APC-conjugated anti-CD3 (clone 17A2, eBioscience),
- B- Spleen: mix 1: Mo: FITC-conjugated anti-CD11b (ThermoFisher Scientific), APC-conjugated anti-Ly-6B.2 (clone 7/4, Biorad), PE-conjugated anti-Ly6G; mix 2: lymphocytes: AF700-conjugated anti-CD45, FITC-conjugated anti-CD4 (clone RM4-5; eBioscience), PercpCy5-conjugated anti-CD8 (clone 53-6.7, BD Pharmingen), APC-conjugated anti-CD3 (clone 17A2, eBioscience),

- C- Blood: mix 1: Mo: FITC-conjugated anti-CD11b (ThermoFisher Scientific), APC-conjugated anti-Ly-6B.2 (clone 7/4, Biorad), PE-conjugated anti-Ly6G; mix 2: lymphocytes: AF700-conjugated anti-CD45, FITC-conjugated anti-CD4 (clone RM4-5; eBioscience), PercpCy5-conjugated anti-CD8 (clone 53-6.7, BD Pharmingen), APC-conjugated anti-CD3 (clone 17A2, eBioscience),
- D- Muscle: mix 1: Mo: AF700-conjugated anti-CD45, FITC-conjugated anti-CD11b (ThermoFisher Scientific), APC-conjugated anti-Ly-6B.2 (clone 7/4, Biorad), PE-conjugated anti-Ly6G; mix 2: lymphocytes: AF700-conjugated anti-CD45, FITC-conjugated anti-CD4 (clone RM4-5; eBioscience), PercpCy5-conjugated anti-CD8 (clone 53-6.7, BD Pharmingen), APC-conjugated anti-CD3 (clone 17A2, eBioscience), mix 3: Macrophages: AF700-conjugated anti-CD45, FITC-conjugated anti-CD11b (ThermoFisher Scientific), PE-conjugated anti-Ly6G, PB-conjugated anti-CD64, APC-conjugated anti-F4/80, PerCP5.5-conjugated anti-MHCII (BD Biosciences), PE-Cyanine7-conjugated anti-CD11c (cloneN418, eBioscience)

Regarding results/discussion:

6. NIR controls are referred as littermates of the irradiated mice. However, there are studies showing the existence of bystander effects from irradiated animals to non-irradiated animals when they are cage mates, especially after high dose RT. The author should address this point in the discussion.

The radiation-induced bystander effect refers to a unique process in which factors released by irradiated cells or tissues exert effects on other parts of the animal not exposed to radiation, causing genomic instability, stress responses and altered apoptosis or cell proliferation^{60 61 62}. The bystander effect can't be transferred from one animal to another animal. Moreover, in this study we didn't mix animal cages of irradiated animals and non-irradiated control littermates.

7. How do the authors explain such an early influx of M-2 like MFs in the muscle? In the discussion they mention the importance of this pro-angiogenic MFs subpopulation in the resolution of Myocardial infarction (MI), however in MI but also in other types of sterile injuries the content of M2-like MFs peaks at 5/7 days post-injury not earlier. Also, because they are gradually differentiating from M1-like Mo/MFs which are the first ones reaching the injured tissue. Author should discuss their findings in relation to this and hypothesize about the reason of the difference between them and what is known in the literature.

As suggested by the reviewer, we clarify below and add in the discussion highlighted in yellow (page 12, line 15-20), that ionizing radiation injury is a different model of sterile injury compared to myocardial infarction. We have an increase of M2-like macrophages at D1 in irradiated muscle. In fact, we suppose that the increase of M2 like macrophages at D1 constitute a **transient adaptative immune**

response to the tissue aggression after radiation exposure. We have several hypotheses, and it could be related to:

- The proliferation of M2 resident cells in the tissue.
- The defect in M2 egress (migration capability)⁶³.
- Resistance of radio induced apoptosis of M2 cells process in the muscle.

8. Furthermore, I would like them to comment on the sustained levels of neutrophils, Mo(hi) and Mo(lo) in the blood (depicted in Figure 2C) and provide a hypothesis or details on why the fluctuations in these cells observed in the spleen and muscle are not reflected in the blood.

As suggested by the reviewer, we clarify below and add in the discussion highlighted in yellow (page 12, line 11-13), that inflammatory cells proliferation, migration and recruitment is a dynamic process. Two organs, bone marrow and spleen, produce innate cells and we can see the proliferation in these. For example, at D1 post-IR there is a proliferation of innate cells, which then go to the blood and are recruited in the muscle. However, the number of innate cells in BM, spleen is around 10^6 and in the blood 10^4 , nevertheless in muscle the number of innate cells is low (25 cells at D1) compared to the level of circulating cells suggesting that there is an accumulation in the blood.

9. Moreover, it would have been great to see how transfer of M2-like MFs can help the healing process after such a high radiation dose. In case the data could not be provided, the author should discuss expected outcomes after such a transfer.

we agree with the reviewer on the importance of the M2 like M ϕ injection to confirm the hypotheses and the role of M ϕ in vascular response for tissue regeneration. However, Macrophage experimentation typically involves *ex vivo* culture (e.g., from the peritoneum), or the differentiation from bone marrow progenitor cells to form bone marrow-derived macrophages. As discussed, macrophage provenance, culture conditions (L929 supernatant quality is critical), and inflammatory stimuli will naturally affect cell phenotype, function, and inflammatory status. Cell heterogeneity is a major limitation of culturing primary macrophages and it is therefore desirable to robustly control conditions *in vitro* in order to fully characterize the response of macrophages to specific and quantifiable stimuli⁵⁰. In any case, the further experiments that we would provide for clinical transfer will be based on the nanoparticle administration to modulate the phenotype of the M ϕ ^{27,51}.

References

- 1 Reyes, E. H. *et al.* Medical Response to Radiological Accidents in Latin America and International Assistance. *Radiation research* **185**, 359-365, doi:10.1667/RR14292.1 (2016).
- 2 Zhao, W. & Robbins, M. E. Inflammation and chronic oxidative stress in radiation-induced late normal tissue injury: therapeutic implications. *Current medicinal chemistry* **16**, 130-143, doi:10.2174/092986709787002790 (2009).
- 3 Bey, E. *et al.* [Radiation burn "innovating therapeutic approach"]. *Annales de chirurgie plastique et esthetique* **55**, 354-362, doi:10.1016/j.anplas.2010.07.009 (2010).
- 4 Lataillade, J. J. *et al.* New approach to radiation burn treatment by dosimetry-guided surgery combined with autologous mesenchymal stem cell therapy. *Regenerative medicine* **2**, 785-794, doi:10.2217/17460751.2.5.785 (2007).
- 5 Muller, K. & Meineke, V. Advances in the management of localized radiation injuries. *Health physics* **98**, 843-850, doi:10.1097/HP.0b013e3181adcb7 (2010).
- 6 Azzam, E. I., Jay-Gerin, J. P. & Pain, D. Ionizing radiation-induced metabolic oxidative stress and prolonged cell injury. *Cancer letters* **327**, 48-60, doi:10.1016/j.canlet.2011.12.012 (2012).
- 7 Kam, W. W. & Banati, R. B. Effects of ionizing radiation on mitochondria. *Free radical biology & medicine* **65**, 607-619, doi:10.1016/j.freeradbiomed.2013.07.024 (2013).
- 8 Tedla, M. *et al.* Impact of radiotherapy on laryngeal intrinsic muscles. *Eur Arch Otorhinolaryngol* **269**, 953-958, doi:10.1007/s00405-011-1686-8 (2012).
- 9 Masuda, S. *et al.* Time- and dose-dependent effects of total-body ionizing radiation on muscle stem cells. *Physiological reports* **3**, doi:10.14814/phy2.12377 (2015).
- 10 Quinlan, J. G. *et al.* Regeneration-blocked mdx muscle: in vivo model for testing treatments. *Muscle & nerve* **20**, 1016-1023, doi:10.1002/(sici)1097-4598(199708)20:8<1016::aid-mus12>3.0.co;2-t (1997).
- 11 Hsu, H. Y., Chai, C. Y. & Lee, M. S. Radiation-induced muscle damage in rats after fractionated high-dose irradiation. *Radiation research* **149**, 482-486 (1998).
- 12 Gerstner, H. B., Lewis, R. B. & Richey, E. O. Early effects of high intensity x-radiation on skeletal muscle. *The Journal of general physiology* **37**, 445-459, doi:10.1085/jgp.37.4.445 (1954).
- 13 Persons, C. C., Hermens, A. F., Franken, N. A. & Wondergem, J. Muscle wasting after radiotherapy in young and adult rats. *Oncology reports* **8**, 1117-1122, doi:10.3892/or.8.5.1117 (2001).
- 14 Sun, W. *et al.* Adipose-Derived Stem Cells Alleviate Radiation-Induced Muscular Fibrosis by Suppressing the Expression of TGF-beta1. *Stem cells international* **2016**, 5638204, doi:10.1155/2016/5638204 (2016).
- 15 Silvestre, J. S. Pro-angiogenic cell-based therapy for the treatment of ischemic cardiovascular diseases. *Thrombosis research* **130 Suppl 1**, S90-94, doi:10.1016/j.thromres.2012.08.287 (2012).
- 16 Billis, W., Fuks, Z. & Kolesnick, R. Signaling in and regulation of ionizing radiation-induced apoptosis in endothelial cells. *Recent progress in hormone research* **53**, 85-92; discussion 93 (1998).
- 17 Zouggari, Y. *et al.* B lymphocytes trigger monocyte mobilization and impair heart function after acute myocardial infarction. *Nature medicine* **19**, 1273-1280, doi:10.1038/nm.3284 (2013).
- 18 Loinard, C. *et al.* Inhibition of prolyl hydroxylase domain proteins promotes therapeutic revascularization. *Circulation* **120**, 50-59, doi:10.1161/CIRCULATIONAHA.108.813303 (2009).
- 19 Loinard, C. *et al.* HuMSC-EV induce monocyte/macrophage mobilization to orchestrate neovascularization in wound healing process following radiation injury. *Cell death discovery* **9**, 38, doi:10.1038/s41420-023-01335-y (2023).

- 20 Jung, K. *et al.* Elevated ARG1 expression in primary monocytes-derived macrophages as a predictor of radiation-induced acute skin toxicities in early breast cancer patients. *Cancer biology & therapy* **16**, 1281-1288, doi:10.1080/15384047.2015.1056945 (2015).
- 21 Handschel, J. *et al.* Increase of RM3/1-positive macrophages in radiation-induced oral mucositis. *The Journal of pathology* **193**, 242-247, doi:10.1002/1096-9896(2000)9999:9999<::AID-PATH754>3.0.CO;2-P (2001).
- 22 Song, C. *et al.* Nanomaterials targeting macrophages in sepsis: A promising approach for sepsis management. *Frontiers in immunology* **13**, 1026173, doi:10.3389/fimmu.2022.1026173 (2022).
- 23 Luo, G. *et al.* Nanoplatfoms for Sepsis Management: Rapid Detection/Warning, Pathogen Elimination and Restoring Immune Homeostasis. *Nano-micro letters* **13**, 88, doi:10.1007/s40820-021-00598-3 (2021).
- 24 Yang, Y. *et al.* Targeted silver nanoparticles for rheumatoid arthritis therapy via macrophage apoptosis and Re-polarization. *Biomaterials* **264**, 120390, doi:10.1016/j.biomaterials.2020.120390 (2021).
- 25 Taratummarat, S. *et al.* Gold nanoparticles attenuates bacterial sepsis in cecal ligation and puncture mouse model through the induction of M2 macrophage polarization. *BMC microbiology* **18**, 85, doi:10.1186/s12866-018-1227-3 (2018).
- 26 Jiang, L. *et al.* Improvement in phenotype homeostasis of macrophages by chitosan nanoparticles and subsequent impacts on liver injury and tumor treatment. *Carbohydrate polymers* **277**, 118891, doi:10.1016/j.carbpol.2021.118891 (2022).
- 27 Kateh Shamshiri, M., Jaafari, M. R. & Badiie, A. Preparation of liposomes containing IFN-gamma and their potentials in cancer immunotherapy: In vitro and in vivo studies in a colon cancer mouse model. *Life sciences* **264**, 118605, doi:10.1016/j.lfs.2020.118605 (2021).
- 28 Cochain, C. *et al.* Regulation of monocyte subset systemic levels by distinct chemokine receptors controls post-ischaemic neovascularization. *Cardiovascular research* **88**, 186-195, doi:10.1093/cvr/cvq153 (2010).
- 29 Nahrendorf, M. *et al.* The healing myocardium sequentially mobilizes two monocyte subsets with divergent and complementary functions. *The Journal of experimental medicine* **204**, 3037-3047, doi:10.1084/jem.20070885 (2007).
- 30 Troidl, C. *et al.* Classically and alternatively activated macrophages contribute to tissue remodelling after myocardial infarction. *Journal of cellular and molecular medicine* **13**, 3485-3496, doi:10.1111/j.1582-4934.2009.00707.x (2009).
- 31 Holler, V. *et al.* Pravastatin limits radiation-induced vascular dysfunction in the skin. *The Journal of investigative dermatology* **129**, 1280-1291, doi:10.1038/jid.2008.360 (2009).
- 32 Silvestre, J. S. *et al.* Vascular endothelial growth factor-B promotes in vivo angiogenesis. *Circulation research* **93**, 114-123, doi:10.1161/01.RES.0000081594.21764.44 (2003).
- 33 Yin, T. *et al.* Genetically modified human placenta-derived mesenchymal stem cells with FGF-2 and PDGF-BB enhance neovascularization in a model of hindlimb ischemia. *Molecular medicine reports* **12**, 5093-5099, doi:10.3892/mmr.2015.4089 (2015).
- 34 Jang, E., Albadawi, H., Watkins, M. T., Edelman, E. R. & Baker, A. B. Syndecan-4 proteoliposomes enhance fibroblast growth factor-2 (FGF-2)-induced proliferation, migration, and neovascularization of ischemic muscle. *Proceedings of the National Academy of Sciences of the United States of America* **109**, 1679-1684, doi:10.1073/pnas.1117885109 (2012).
- 35 Skora, J. P. *et al.* Local intramuscular administration of ANG1 and VEGF genes using plasmid vectors mobilizes CD34+ cells to peripheral tissues and promotes angiogenesis in an animal model. *Biomedicine & pharmacotherapy = Biomedecine & pharmacotherapie* **143**, 112186, doi:10.1016/j.biopha.2021.112186 (2021).
- 36 Yu, B. *et al.* Exosomes secreted from GATA-4 overexpressing mesenchymal stem cells serve as a reservoir of anti-apoptotic microRNAs for cardioprotection. *International journal of cardiology* **182**, 349-360, doi:10.1016/j.ijcard.2014.12.043 (2015).

- 37 Teng, X. *et al.* Mesenchymal Stem Cell-Derived Exosomes Improve the Microenvironment of Infarcted Myocardium Contributing to Angiogenesis and Anti-Inflammation. *Cellular physiology and biochemistry : international journal of experimental cellular physiology, biochemistry, and pharmacology* **37**, 2415-2424, doi:10.1159/000438594 (2015).
- 38 Zhang, Z. *et al.* Pretreatment of Cardiac Stem Cells With Exosomes Derived From Mesenchymal Stem Cells Enhances Myocardial Repair. *Journal of the American Heart Association* **5**, doi:10.1161/JAHA.115.002856 (2016).
- 39 Contreras-Shannon, V. *et al.* Fat accumulation with altered inflammation and regeneration in skeletal muscle of CCR2-/- mice following ischemic injury. *American journal of physiology. Cell physiology* **292**, C953-967, doi:10.1152/ajpcell.00154.2006 (2007).
- 40 Arnold, L. *et al.* CX3CR1 deficiency promotes muscle repair and regeneration by enhancing macrophage ApoE production. *Nature communications* **6**, 8972, doi:10.1038/ncomms9972 (2015).
- 41 Arnold, L. *et al.* Inflammatory monocytes recruited after skeletal muscle injury switch into antiinflammatory macrophages to support myogenesis. *The Journal of experimental medicine* **204**, 1057-1069, doi:10.1084/jem.20070075 (2007).
- 42 Martinez, C. O. *et al.* Regulation of skeletal muscle regeneration by CCR2-activating chemokines is directly related to macrophage recruitment. *American journal of physiology. Regulatory, integrative and comparative physiology* **299**, R832-842, doi:10.1152/ajpregu.00797.2009 (2010).
- 43 Sun, D. *et al.* Bone marrow-derived cell regulation of skeletal muscle regeneration. *FASEB journal : official publication of the Federation of American Societies for Experimental Biology* **23**, 382-395, doi:10.1096/fj.07-095901 (2009).
- 44 Lu, H. *et al.* Macrophages recruited via CCR2 produce insulin-like growth factor-1 to repair acute skeletal muscle injury. *FASEB journal : official publication of the Federation of American Societies for Experimental Biology* **25**, 358-369, doi:10.1096/fj.10-171579 (2011).
- 45 Shintani, S. *et al.* Augmentation of postnatal neovascularization with autologous bone marrow transplantation. *Circulation* **103**, 897-903, doi:10.1161/01.cir.103.6.897 (2001).
- 46 Fuchs, S. *et al.* Transendocardial delivery of autologous bone marrow enhances collateral perfusion and regional function in pigs with chronic experimental myocardial ischemia. *Journal of the American College of Cardiology* **37**, 1726-1732, doi:10.1016/s0735-1097(01)01200-1 (2001).
- 47 Kalka, C. *et al.* Transplantation of ex vivo expanded endothelial progenitor cells for therapeutic neovascularization. *Proceedings of the National Academy of Sciences of the United States of America* **97**, 3422-3427, doi:10.1073/pnas.97.7.3422 (2000).
- 48 Beck, H. *et al.* Participation of bone marrow-derived cells in long-term repair processes after experimental stroke. *Journal of cerebral blood flow and metabolism : official journal of the International Society of Cerebral Blood Flow and Metabolism* **23**, 709-717, doi:10.1097/01.WCB.0000065940.18332.8D (2003).
- 49 Ziegelhoeffer, T. *et al.* Bone marrow-derived cells do not incorporate into the adult growing vasculature. *Circulation research* **94**, 230-238, doi:10.1161/01.RES.0000110419.50982.1C (2004).
- 50 Wynn, T. A., Chawla, A. & Pollard, J. W. Macrophage biology in development, homeostasis and disease. *Nature* **496**, 445-455, doi:10.1038/nature12034 (2013).
- 51 Tian, M. *et al.* Adipose-Derived Biogenic Nanoparticles for Suppression of Inflammation. *Small* **16**, e1904064, doi:10.1002/smll.201904064 (2020).
- 52 Knowlton, A. A. & Lee, A. R. Estrogen and the cardiovascular system. *Pharmacology & therapeutics* **135**, 54-70, doi:10.1016/j.pharmthera.2012.03.007 (2012).
- 53 Crescioli, C. The Role of Estrogens and Vitamin D in Cardiomyocyte Protection: A Female Perspective. *Biomolecules* **11**, doi:10.3390/biom11121815 (2021).

- 54 Kassi, E. *et al.* Vascular Inflammation and Atherosclerosis: The Role of Estrogen Receptors. *Current medicinal chemistry* **22**, 2651-2665, doi:10.2174/0929867322666150608093607 (2015).
- 55 Hajjalizadeh, Z. & Khaksari, M. The protective effects of 17-beta estradiol and SIRT1 against cardiac hypertrophy: a review. *Heart failure reviews* **27**, 725-738, doi:10.1007/s10741-021-10171-0 (2022).
- 56 Wang, M., Tsai, B. M., Reiger, K. M., Brown, J. W. & Meldrum, D. R. 17-beta-Estradiol decreases p38 MAPK-mediated myocardial inflammation and dysfunction following acute ischemia. *Journal of molecular and cellular cardiology* **40**, 205-212, doi:10.1016/j.yjmcc.2005.06.019 (2006).
- 57 Kim, Y. D. *et al.* 17 beta-Estradiol prevents dysfunction of canine coronary endothelium and myocardium and reperfusion arrhythmias after brief ischemia/reperfusion. *Circulation* **94**, 2901-2908, doi:10.1161/01.cir.94.11.2901 (1996).
- 58 Zhai, P. *et al.* Effect of estrogen on global myocardial ischemia-reperfusion injury in female rats. *American journal of physiology. Heart and circulatory physiology* **279**, H2766-2775, doi:10.1152/ajpheart.2000.279.6.H2766 (2000).
- 59 Wang, M. *et al.* Mitochondrial connexin 43 in sex-dependent myocardial responses and estrogen-mediated cardiac protection following acute ischemia/reperfusion injury. *Basic research in cardiology* **115**, 1, doi:10.1007/s00395-019-0759-5 (2019).
- 60 Mothersill, C. & Seymour, C. Radiation-induced bystander effects: past history and future directions. *Radiation research* **155**, 759-767, doi:10.1667/0033-7587(2001)155[0759:ribeph]2.0.co;2 (2001).
- 61 Prise, K. M. & O'Sullivan, J. M. Radiation-induced bystander signalling in cancer therapy. *Nature reviews. Cancer* **9**, 351-360, doi:10.1038/nrc2603 (2009).
- 62 Rzeszowska-Wolny, J., Przybyszewski, W. M. & Widel, M. Ionizing radiation-induced bystander effects, potential targets for modulation of radiotherapy. *European journal of pharmacology* **625**, 156-164, doi:10.1016/j.ejphar.2009.07.028 (2009).
- 63 Kosco, B. *et al.* Gut-resident CX3CR1(hi) macrophages induce tertiary lymphoid structures and IgA response in situ. *Science immunology* **5**, doi:10.1126/sciimmunol.aax0062 (2020).

REVIEWERS' COMMENTS:

Reviewer #1 (Remarks to the Author):

The authors have revised the manuscript in accordance with comments. The revised version has been found acceptable for the publication.